# Cryo-EM structures of the human P2X1 receptor reveal subtype-specific architecture and antagonism by supramolecular ligand-binding

Adam C. Oken [1,3], Nicolas E. Lisi[1,3], Ismayn A. Ditter [1,3], Haoyuan Shi [1,3], Nadia A. Nechiporuk[1] & Steven E. Mansoor [1,2] ✉

P2X receptors are a family of seven trimeric non-selective cation channels that are activated by extracellular ATP to play roles in the cardiovascular, neuronal, and immune systems. Although it is known that the P2X1 receptor subtype has increased sensitivity to ATP and fast desensitization kinetics, an underlying molecular explanation for these subtype-selective features is lacking. Here we report high-resolution cryo-EM structures of the human P2X1 receptor in the apo closed, ATP-bound desensitized, and the high-affinity antagonist NF449-bound inhibited states. The apo closed and ATP-bound desensitized state structures of human P2X1 define subtype-specific properties such as distinct pore architecture and ATP-interacting residues. The NF449-bound inhibited state structure of human P2X1 reveals that NF449 has a unique dual-ligand supramolecular binding mode at the interface of neighboring protomers, inhibiting channel activation by overlapping with the canonical P2X receptor ATP-binding site. Altogether, these data define the molecular pharmacology of the human P2X1 receptor laying the foundation for structure-based drug design.

Therapy to inhibit platelet activation is fundamental for the treatment of atherosclerotic cardiovascular and cerebrovascular disease[1]. Following injury to the walls of blood vessels, platelets adhere to the newly exposed sub-endothelium, become activated, and release mediators such as thromboxane A2 and purine nucleotides that act to further amplify platelet activation at the site of vessel injury[2]. The released adenosine nucleotides modulate platelet activation through two classes of purinergic (P2) receptors: the family of seven ATP-gated P2X receptor (P2XR) ion channels (denoted P2X1 - P2X7) and the family of eight G-protein coupled P2Y receptors that recognize both extracellular ATP and extracellular ADP[3–8]. As such, P2X and P2Y receptors represent targets for antithrombotic therapy[9–11].

The action of ADP on platelets through $P2Y_{12}$ receptors is well documented and its inhibition has had tremendous clinical impact with the use of antiplatelet agents such as clopidogrel[12]. However, that benefit has always been tempered by the clear and demonstrable increased risk of bleeding that occurs with $P2Y_{12}$ receptor antagonism[13]. On the other hand, the less studied actions of extracellular ATP on platelets mediated through the P2X1 receptor is a potential area of therapeutic investigation to inhibit platelet activation at sites where shear forces are high while mitigating hemorrhagic risk by not affecting platelets under conditions of normal hemostasis[14]. Indeed, P2X1-deficient mice do not have a prolongation in their bleeding time relative to wild-type mice but do possess resistance to

[1]Department of Chemical Physiology & Biochemistry, Oregon Health & Science University, Portland, OR 97239, USA. [2]Division of Cardiovascular Medicine, Knight Cardiovascular Institute, Oregon Health & Science University, Portland, OR 97239, USA. [3]These authors contributed equally: Adam C. Oken, Nicolas E. Lisi, Ismayn A. Ditter, Haoyuan Shi. ✉e-mail: mansoors@ohsu.edu

systemic thromboembolism following laser-induced injury to blood vessel walls[14]. In addition, the P2X1 receptor antagonist NF449 has an inhibitory effect on platelet activation ex vivo as well as an inhibitory effect on thrombosis in vivo[15]. A molecular explanation for this antagonism has not been defined. Taken together, these findings suggest that pharmacological therapy to modify platelet activation by targeting antagonism at the P2X1 receptor should be investigated.

Currently, there are no available structures for several of the P2XR subtypes, including the P2X1 receptor, and only one structure for a human ortholog (the human P2X3 receptor), leaving unanswered questions regarding the pharmacological and functional differences between them[16]. For example, it is not fully understood why P2X1 and P2X3 receptors are more sensitive to ATP activation than either P2X2 or P2X4 receptors or why the P2X7 receptor is the least sensitive to ATP[17]. In addition, the kinetics of ion channel gating in response to agonists vary dramatically between P2XR subtypes, with P2X1 and P2X3 receptors undergoing rapid desensitization (milliseconds), P2X2, P2X4, and P2X5 receptors having slow rates of desensitization (seconds), and the P2X7 receptor showing no observable desensitization[5,18–22].

Within the P2X family of ion channels, the functional and pharmacological properties of the P2X1 receptor most closely mirror that of the P2X3 receptor[5]. Yet, critical differences in the molecular pharmacology of each subtype remain poorly understood. For example, the small-molecule AF-219 can antagonize the P2X3 receptor with nanomolar affinity yet is three orders of magnitude less effective at the P2X1 receptor[23]. Similarly, the small-molecule NF449 is significantly more effective at antagonizing the P2X1 receptor than it is at the P2X3 receptor[24]. Without high-resolution structures of each specific P2XR subtype, especially for human orthologs, methods for using rational drug design to facilitate the development of selective agonist and antagonist ligands remain suboptimal. A molecular examination for the unique structural features of the P2X1 receptor alongside close comparison to the functionally similar P2X3 receptor could be the key to unlocking its pharmacological potential.

Herein we use single particle cryogenic electron microscopy (cryo-EM) to study the structural features governing human P2X1 receptor (hP2X1) function in the apo closed and the ATP-bound desensitized states, as well as its inhibition by the high-affinity antagonist NF449. Comparison between high-resolution structures of hP2X1 in multiple conformations with structures of the well-studied human P2X3 receptor (hP2X3) in the same conformations helps draw conclusions about unique structure/function relationships that exist within this P2XR subtype. Further, the identification of a supramolecular antagonist ligand-binding site in hP2X1 will facilitate the development of potent and specific molecules for therapeutic intervention.

## Results

### Overall architecture of the hP2X1 receptor

We used single particle cryo-EM to determine high-resolution structures of the full-length wild-type hP2X1 receptor in the apo closed (Fig. 1) and ATP-bound desensitized (Fig. 2) states (Supplementary Figs. 1, 2, Supplementary Table 1). To identify subtype-specific features

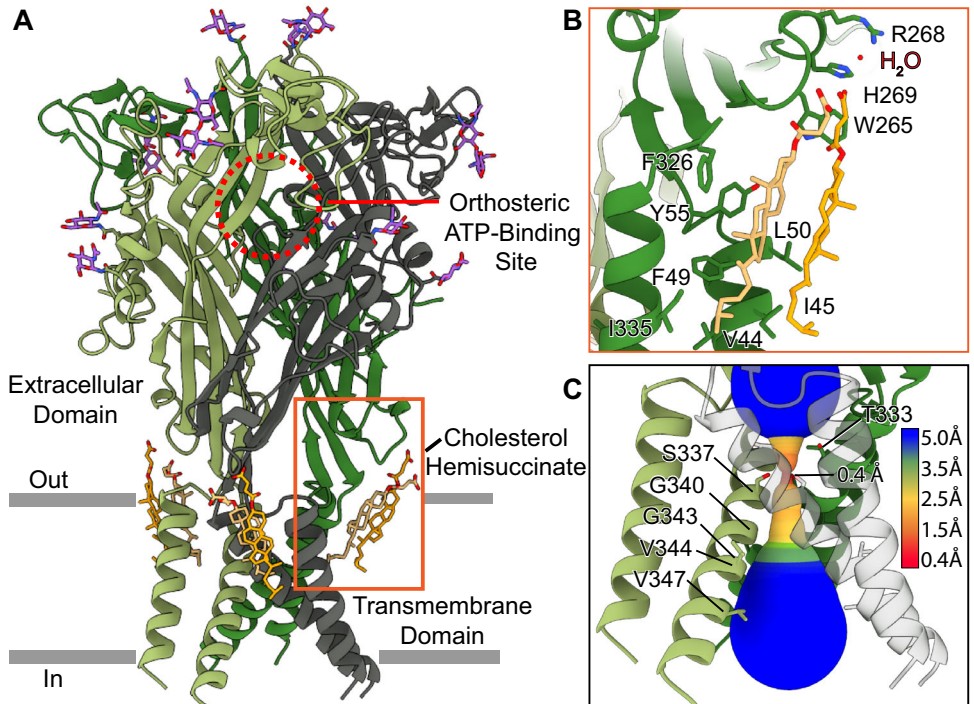

**Fig. 1 | Architecture and pore structure of hP2X1 in the apo closed state.**
**A** Ribbon representation of the hP2X1 receptor in the apo closed state, highlighting one of the three empty orthosteric ATP-binding sites (outlined in a red circle), a moderate amount of glycosylation sites in the extracellular domain (N-acetylglucosamine molecules colored in purple), and two cholesterol hemisuccinate (CHS) molecules (shown in tan and orange) bound per protomer. Each protomer of the receptor is colored differently (green, olive, and dark gray). **B** Magnified and 40° rotated view of the two CHS molecules (tan and orange) highlighting the key interactions at the interface between TM1 and TM2 of one protomer. The inner CHS molecule (tan) interacts with the sidechains of V44, I45, F49, L50, Y55 on TM1 as well as F326 and I335 on TM2. The outer CHS molecule (orange) interacts with the sidechains of I45 and L50 on TM1 as well as the inner CHS. The succinate group of the outer CHS molecule hydrogen bonds with the sidechains of residues W265 and H269 (3.5 Å and 2.9 Å, respectively) located on the lower body as well as a water molecule (3.3 Å). **C** The ion permeation pathway of hP2X1 in the apo closed state reveals that the constriction gate of the closed pore at its narrowest point is 0.4 Å, created by the sidechains of three symmetry-related S337 residues on TM2. The constriction gate is only 7 Å deep into the plasma membrane. Due to the shallow gate, the overall shape of the transmembrane domain in the apo closed state conformation resembles a teepee. For the pore size plot, different colors represent different radii, as generated by the program MOLEonline[60]. While V344 does not line the pore or contribute to the conduction path of hP2X1 in the apo closed state, it is shown here as a reference to contrast its critical role as the residue defining the gate in the ATP-bound desensitized state (Fig. 2).

of the hP2X1 receptor, structures of hP2X1 are compared to the equivalent conformational states of hP2X3 (Figs. 3, 4)[16]. In addition, a high-resolution structure of hP2X1 bound to the selective and high-affinity antagonist, NF449, reveals a unique dual-ligand supramolecular binding mode at a competitive ligand-binding site not previously observed for any other P2XR antagonist (Fig. 5, Supplementary Fig. 1, 2, Supplementary Table 1).

Like other P2XR subtypes, hP2X1 exhibits a trimeric architecture (Fig. 1A, Supplementary Fig. 3A, B). Each protomer, described to resemble a breaching dolphin, is composed of a large, hydrophilic extracellular domain (ECD), two transmembrane (TM) spanning alpha helices (the outer TM1 and pore-lining TM2), and intracellular N- and C- termini (Supplementary Fig. 3A, B). The protomers are woven together via extensive domain swapping, featuring a 120° clockwise rotation between each protomer, from the ECD relative to the transmembrane domain (TMD) (Supplementary Fig. 3C). While the global architecture of hP2X1 is similar to other P2XRs, there are distinct structural features that will be discussed in greater detail.

## Architecture and structural properties of hP2X1 in the apo closed state

Our high-resolution cryo-EM structure of full-length wild-type hP2X1 in the apo closed state at 2.7 Å resolution provides key insights into the subtype-specific features of this receptor (Fig. 1, Supplementary Fig. 1, 2, Supplementary Table 1). An apo closed state conformation was isolated following the purification protocol outlined in the Methods. Briefly, mammalian cells overexpressing hP2X1 were incubated with saturating concentrations of NF449 antagonist prior to lysis to prevent binding of endogenous ATP, followed by extensive dialysis of cellular membranes to remove antagonist before detergent solubilization and reconstitution for purification. As expected, the apo state structure of hP2X1 has an empty ligand-binding pocket and a closed pore (Fig. 1A, C). The pore lumen is lined with residues T333, S337, G340, G343, and V347 on TM2 (Fig. 1C). Residue S337 from each protomer defines the narrowest region of the gate in the apo closed state, constricting the pore to a radius of 0.4 Å, which is too narrow to pass dehydrated Na$^+$ ions (Fig. 1C)[25]. The architecture of the TMD is a key feature of the apo closed state: the pore lining TM2 helices (one TM2 helix from each protomer) are arranged to form a closed channel with the constriction gate (residue S337 from each protomer) near the extracellular leaflet of the lipid bilayer, only 7 Å deep into the plasma membrane, as measured from the start of TM2 (residue I328). As a result of this "shallow" gate, the overall shape of the TMD in the apo closed state structure resembles a "teepee" (Fig. 1A, C)[26,27].

The addition of cholesterol hemi-succinate (CHS) during the purification was necessary to stabilize the hP2X1 receptor for structural determination, as the initial purifications performed without the addition of CHS resulted in cryo-EM reconstructions lacking density for the TMD. Consequently, we observed density for and modeled two CHS molecules bound per protomer in the TMD of hP2X1 (Fig. 1A, B, Supplementary Fig. 4). The two CHS molecules occupy a cavity between TM1 and TM2, at the interface between the ECD and the TMD, located on what would be the extracellular leaflet of the lipid bilayer (Fig. 1A, B, Supplementary Fig. 4). The CHS molecules are stacked on top of each other, making numerous hydrophobic contacts, with the inner CHS interacting predominantly with the receptor and the outer CHS making extensive interactions with the inner CHS molecule. The hydrocarbon sidechain of the inner CHS makes hydrophobic contacts with the sidechains of V44, I45 and F49 on TM1 as well as I335 on TM2; the steroid nucleus contacts F326 on TM2 as well as L50 and Y55 on TM1; the succinate group is too far from the receptor to make any hydrogen bonding interactions (Fig. 1B, Supplementary Fig. 4A–C). The hydrocarbon sidechain of the outer CHS makes hydrophobic interactions with the first CHS molecule and the sidechains of residues

I45 and L50 on TM1 (Fig. 1B, Supplementary Fig. 4D–F). The steroid nucleus of the outer CHS molecule makes hydrophobic interactions with the steroid nucleus of the inner CHS (Fig. 1B). Finally, the succinate group of the outer CHS molecule creates hydrogen bonding interactions with the sidechains of residues W265 and H269 (3.5 Å and 2.9 Å, respectively) located on the lower body domain of the receptor (Fig. 1B, Supplementary Fig. 3A, 4D–F). The succinate group of the outer CHS also forms a hydrogen bond (3.3 Å) with a water molecule, which is stabilized by interactions with H266, R268, and H269 (Supplementary Fig. 4D–F). Another interesting observation is that residues I45, F49, and L50 are located on a short stretch of a π-helix on TM1 (residues 45-52), which is initiated and terminated by two glycine residues on either end of the non-standard helix (G46 and G54). This π-helix helps position the sidechains of I45, F49, and L50 to interact with the CHS molecules. The mutation L50A does not significantly impact current responses to ATP (Supplementary Fig. 5). On the other hand, mutations I45A, F49A, and Y55A impact function of the receptor relative to wild-type hP2X1, significantly decreasing current responses to applications of ATP, suggesting these residues potentially play a role to stabilize the receptor's TMD in the plasma membrane (Supplementary Fig. 5).

Overall, the apo closed state structure of the hP2X1 receptor resembles the apo closed state structure of the hP2X3 receptor, with RMSD's of 0.8 Å across pruned atom pairs and 3.1 Å across all atom pairs (Fig. 3A)[28]. While the architecture of the TMD for both P2XR subtypes maintains the teepee-like tertiary structure in the apo closed state, the TMs of hP2X1 and hP2X3 have notable differences in orientation. Compared to hP2X3, TM1 of hP2X1 is rotated inward by ~19°, with the center of rotation located near the extracellular end of the helix (Fig. 3B). For TM2, a smaller change in orientation is observed, with the helix in hP2X1 rotated inward by ~14° (Fig. 3B). However, this difference in helical pitch for TM2 between hP2X1 and hP2X3 results in key differences to the molecular architecture of each pore, specifically altering the residues that define the constriction gate. For hP2X3, the closed pore is defined by a series of three sets of residues from each protomer: I323 (pore radius of 0.4 Å), followed by V326 (pore radius of 0.8 Å), and then T330 (pore radius of 0.8 Å)[16]. In comparison, the pore of hP2X1 in the apo closed state has an initial narrowing at T333 (pore radius of 0.8 Å) that tapers down to a single constriction gate set by S337 from each protomer (pore radius of 0.4 Å) (Fig. 1C). The gate in the apo closed state of hP2X1 (at residue S337) occurs at the same register of TM2 as the apo closed state of rat P2X7 (rP2X7) but at a distinct position compared to both hP2X3 and zebrafish P2X4 (zfP2X4)[16,29,30]. The residue responsible for the primary constriction gate of hP2X1 (S337) is one full helical turn or 4.4 Å deeper into the membrane bilayer on TM2 than the gate of hP2X3 (I323) (Fig. 3C). The most structurally analogous residue to S337 from hP2X1 is A327 in hP2X3. However, A327 appears to play no significant role in the conductance pathway of hP2X3[16]. The molecular architecture of hP2X1 in the apo closed state provides insight into structural differences between subtypes as well as unoccupied ligand-binding sites for use in structure-based drug design.

## Architecture and structural properties of hP2X1 in the ATP-bound desensitized state

The cryo-EM structure of hP2X1 in the ATP-bound desensitized state at 2.4 Å resolution provides insight into the pore architecture of the desensitized state conformation and reveals subtype-specific interactions in the P2X1 receptor's orthosteric ATP-binding site (Figs. 2, 4, Supplementary Figs. 1, 2, Supplementary Table 1). This structure is defined as the desensitized state conformation because ATP is bound in the orthosteric pocket and the pore is uniquely closed. The ATP-bound desensitized state conformation was isolated following purification of hP2X1 without the addition of exogenous ATP, as endogenous ATP released during cell lysis binds to the receptor and

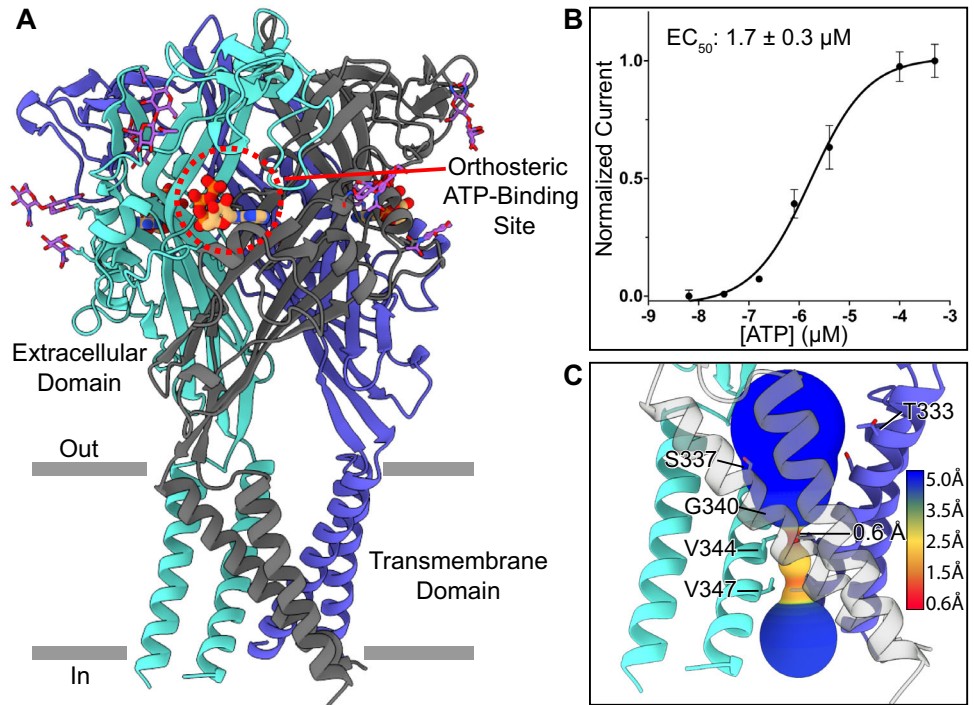

**Fig. 2 | Architecture and pore structure of hP2X1 in the ATP-bound desensitized state. A** Ribbon representation of the hP2X1 receptor in the ATP-bound desensitized state, highlighting one of the three occupied orthosteric ATP-binding sites (outlined in a red circle). There is a moderate amount of glycosylation sites in the extracellular domain (N-acetylglucosamine molecules colored in purple). Each protomer of the receptor is colored differently (blue, cyan, and dark gray). **B** Dose response curve ($EC_{50}$) for wild-type hP2X1 activated by ATP is measured to be $1.7 \pm 0.3\,\mu M$ by two-electrode voltage clamp (TEVC) in *Xenopus laevis* oocytes. The Y-axis describes currents normalized to the current evoked by the initial concentration of ATP applied to each oocyte (100 μM). The reported $EC_{50}$ and error bars represent the mean and standard deviation across triplicate experiments, respectively. **C** The ion permeation pathway of hP2X1 in the ATP-bound desensitized state reveals that the constriction gate of the closed pore at its narrowest point is 0.6 Å, created by the sidechains of three symmetry-related V344 residues on TM2. The constriction gate in the ATP-bound desensitized state structure is 16 Å deep into the plasma membrane, a full 9 Å deeper than the constriction gate of the apo closed state. Due to the deeper gate, the overall shape of the transmembrane domain in the ATP-bound desensitized state conformation resembles a cone. For the pore size plot, different colors represent different radii, as generated by the program MOLEonline[60].

remains bound throughout the purification process and structural analysis. This finding was observed previously for the hP2X3 receptor and is the reason that extra steps were required to successfully purify the apo closed state conformation (see Methods)[16].

The TMD and pore architecture of the ATP-bound desensitized state structure of hP2X1 differs dramatically compared to the closed pore of the apo state (Figs. 1A, C, 2A, C, Supplementary Fig. 6). The conformational changes that take place when the hP2X3 receptor progresses from the apo to open to desensitized state conformations have been described in detail in the helical recoil model of P2XR desensitization[16,27]. The structural changes between the apo closed state and ATP-bound desensitized state of hP2X1 follow a similar form. Briefly, TM1 in the ATP-bound desensitized state of hP2X1, relative to the apo closed state, is rotated downward in-plane by ~5° with the hinge at the extracellular end of TM1 and TM2 is rotated downward in-plane by ~4° with the hinge at the extracellular end of TM2 (Supplementary Fig. 6). These changes are less than the same measurements in hP2X3 because the constriction gate of hP2X1 in the apo closed state is one helical turn more cytoplasmic, creating smaller differences between the two conformations. However, the result is the same, as the movements in the TMD induce an inward rotation of residue V344 on TM2 of hP2X1 (residue V334 on TM2 of hP2X3) to block the pore and form the constriction gate for the ATP-bound desensitized state. The position of the closed gate in hP2X1 is now 16 Å deep into the plasma membrane, as measured from the beginning of the TM2 helix (residue I328). With the gate deeper into the bilayer and both TMs rotated downward, the TMD architecture of hP2X1 in the ATP-bound desensitized state now resembles a "cone" (Fig. 2A, C)[26,27].

The positions of the TMDs between the ATP-bound desensitized states of hP2X1 and hP2X3 are quite similar (Fig. 3D). Compared to hP2X3, TM1 of hP2X1 is rotated downward in-plane by ~8°, with the hinge at the extracellular end of the helix (Fig. 3E). For TM2, a smaller difference in orientation is observed, where the hP2X1 helix is rotated downward in-plane by ~6° with the hinge at the extracellular end of the helix (Fig. 3E). These small rotational differences translate to only small changes in molecular architecture between the desensitized state pores of hP2X1 and hP2X3. The constriction gate in the ATP-bound desensitized state structure of hP2X3 is formed by V334 from each protomer (pore radius of 1.6 Å) with no other residues contributing to significant constriction[16]. The main constriction gate of hP2X1 in the ATP-bound desensitized state is formed by V344 from each protomer (pore radius of 0.6 Å) (Fig. 2C). There is a second discrete gate in the ATP-bound desensitized state of hP2X1 that occurs at residue V347 (pore radius of 1.1 Å) (Fig. 2C). Importantly, the respective constriction gates in the ATP-bound desensitized states of hP2X1 and hP2X3 are structurally located at the same position in TM2 and both are formed by valine residues from each protomer (Fig. 3F). While both pores are unable to pass partially hydrated Na+ ions, the diameter of the constriction gate of hP2X3 in the ATP-bound desensitized state is ~2.5x larger than the diameter of the gate in the ATP-bound desensitized state of hP2X1 (0.6 Å radius in hP2X1 vs. 1.6 Å radius in hP2X3).

### The orthosteric ATP-binding pocket of hP2X1 has a unique composition

Relative to other P2XR subtypes, fast-desensitizing P2XRs are more sensitive to ATP activation[17,18]. In our hands, full-length wild-type hP2X1

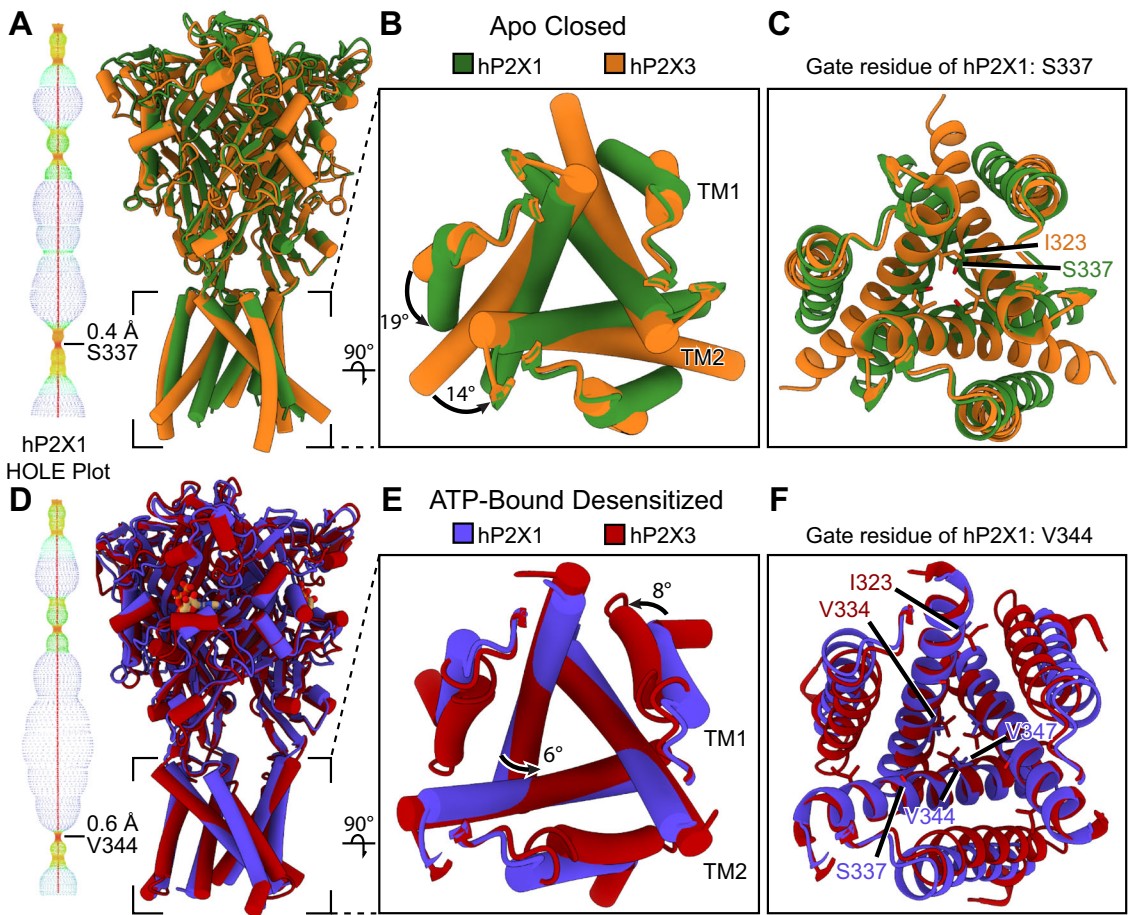

**Fig. 3 | Comparison of overall architecture and pore structure of the hP2X1 receptor vs. the hP2X3 receptor. A–C** Overlayed apo closed state structures of hP2X1 (green) and hP2X3 (orange, PDB code: 5SVJ) highlighting the differences between the two P2XR subtypes. **A** Entire view of both receptor subtypes. To the left of the overlayed structures, a HOLE plot generated of hP2X1 from the apo closed state structure highlights the constriction gate formed by S337 from each protomer[61]. **B** Top-down view of the TMD showing the differences in helical pitch between hP2X1 (green) and hP2X3 (orange) in the apo closed state. **C** Same view as panel B detailing which residues form the constriction site in the apo closed state of hP2X1 compared to hP2X3. Importantly, the constriction gate of hP2X1 in the apo closed state is 7 Å deep into the plasma membrane while the constriction gate of hP2X3 in the apo closed state is only 2.6 Å deep. **D–F** Overlayed ATP-bound desensitized state structures of hP2X1 (blue) and hP2X3 (red, PDB code: 5SVL)

highlighting the differences between the two receptor subtypes. **D** Entire view of both receptor subtypes. To the left of the overlayed structures, a HOLE plot generated of hP2X1 from the ATP-bound desensitized state structure highlights the constriction gate formed by V344 from each protomer[61]. **E** Top-down view of the TMD showing the differences in helical pitch between hP2X1 (blue) and hP2X3 (red) in the ATP-bound desensitized state. **F** Same view as panel E detailing which residues form the constriction site in the ATP-bound desensitized state of hP2X1 compared to hP2X3. The main constriction gate of hP2X1 in the ATP-bound desensitized state is 16 Å deep into the plasma membrane with a secondary constriction site formed by V347 from each protomer. Similarly, the main constriction gate of hP2X3 in the ATP-bound desensitized state is 15 Å deep into the plasma membrane. **A, D** Colors within the HOLE plots represent different pore radii, with red < 1.15 Å, green 1.15 Å < pore radius < 2.30 Å, and blue > 2.30 Å.

has an apparent affinity ($EC_{50}$) of $1.7 \pm 0.3\,\mu M$, as assessed in two-electrode voltage clamp experiments (TEVC) (Fig. 2B). The apparent affinity of the other fast desensitizing P2XR subtype, P2X3, has a similar $EC_{50}$ for ATP which we measured in hP2X3 to be $0.8 \pm 0.1\,\mu M$ by TEVC (Supplementary Fig. 7A). In stark contrast, the apparent affinity of P2X7 receptor for ATP has been reported to be > 100 $\mu M$ and even > 1 mM under certain conditions[17,18]. Despite the broad range of sensitivities to ATP across P2XR subtypes, the orthosteric ATP-binding site, positioned at the interface between adjacent protomers, is highly conserved[26,31]. But, there must be subtype-specific differences that account for the large variance in P2XR sensitivity to extracellular ATP.

The residue composition of the orthosteric ATP-binding pockets between hP2X1 and hP2X3 are similar (Fig. 4A, B). However, there are several differences worth noting. In the ATP-bound desensitized state of hP2X1, ATP is coordinated by seven interactions that are conserved amongst all P2XRs: N290, T186, R292, K68, K70, K190, and K309 (Fig. 4C, D)[26]. In addition to these conserved interactions, there are several unique contacts between ATP and hP2X1. In the analogous

position to P128 in hP2X3, K140 in hP2X1 directly contacts the 2'-hydroxyl on the ribose ring of ATP (2.6 Å) (Fig. 4C, D, Supplementary Fig. 8). Surprisingly, the mutation of K140 to an alanine did not impact the apparent affinity of ATP at hP2X1 as measured by TEVC (Supplementary Fig. 7B). The next direct interaction with ATP that differs between the two subtypes is a substitution of I215 in hP2X3 to V229 in hP2X1, decreasing the size of a small hydrophobic patch below the adenosine base (Supplementary Fig. 8). In addition, there are also residue substitutions that make the pocket of hP2X1 more hydrophilic. For example, T235 in hP2X3 is equivalent to K249 in hP2X1, which extends closer to the orthosteric binding site (Supplementary Fig. 8). The mutation K249A moderately increases the apparent affinity of ATP to hP2X1 (Supplementary Fig. 7B). The presence of a positive charge just outside the orthosteric site might hinder entry of ATP into the orthosteric binding site. Finally, we observed greater apparent cooperativity in activation of hP2X3 compared to hP2X1, with a Hill slope of $2.5 \pm 0.6$ compared to $0.9 \pm 0.2$, respectively (Supplementary Fig. 7A). Since our mutational studies of residues in the orthosteric pocket of

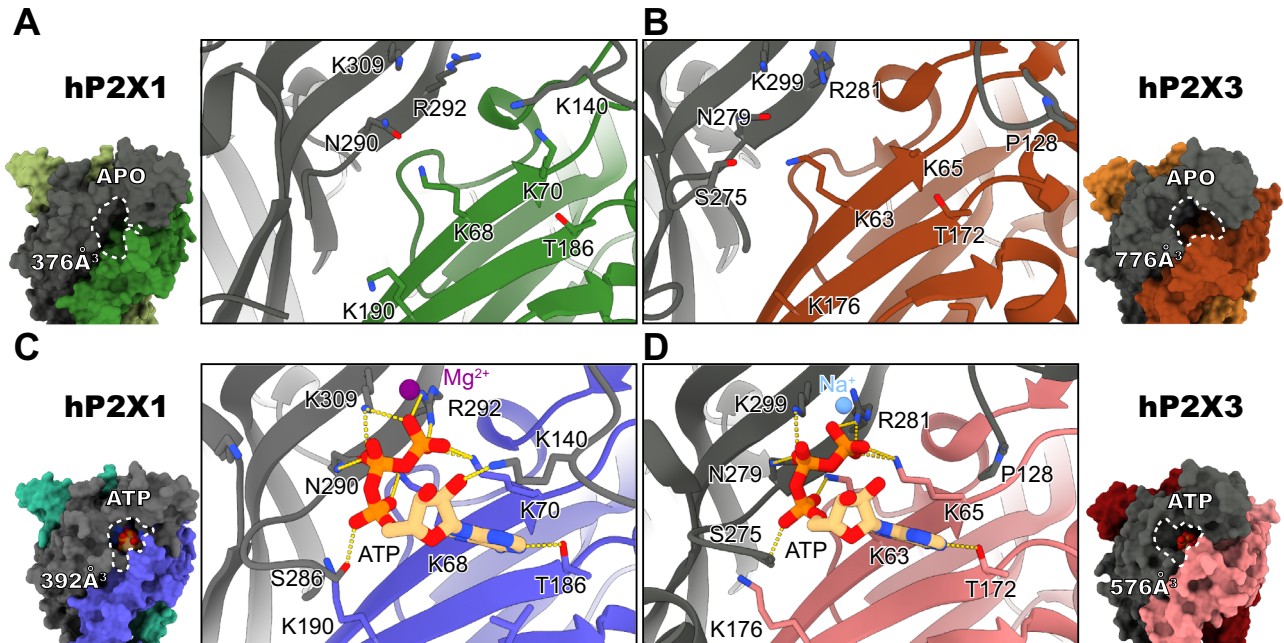

**Fig. 4 | Subtype-specific features of the orthosteric ATP-binding site in hP2X1 compared to hP2X3. A, B** The unoccupied orthosteric ATP-binding site in the apo closed states of hP2X1 (**A**, shades of green and gray) and hP2X3 (**B**, shades of red and gray, PDB code: 5SVJ). The empty orthosteric pocket of hP2X1 at 376 Å³ (**A**) is much smaller than the empty orthosteric pocket found in hP2X3 at 776 Å³ (**B**) as calculated with CASTp (using a 2 Å probe radius)[62]. The empty orthosteric pockets are outlined by a white dotted line. **C, D** ATP is bound to the orthosteric pockets of hP2X1 (**C**, shades of blue and gray) and hP2X3 (**D**, shades of pink and gray, PDB code: 5SVL) in the ATP-bound desensitized states. Between the two fast-desensitizing subtypes, most residues within the orthosteric pocket that interact

with ATP are conserved. However, there are interactions that differ. The location of P128 in hP2X3 is equivalent to K140 in hP2X1, which now interacts with ATP. While hP2X1 has a Mg²⁺ ion that interacts with the γ-phosphate of ATP, there are structures of hP2X3 with Na⁺, Mg²⁺, or Ca²⁺ ions present in the orthosteric pocket[16,34]. **A–D** When comparing the apo closed state to the ATP-bound desensitized state, we observe that the orthosteric pocket in hP2X1 doesn't change much in volume (376 Å³ vs. 392 Å³) while the orthosteric pocket of hP2X3 shrinks significantly (776 Å³ vs. 576 Å³) as calculated with CASTp (using a 2 Å probe radius)[62]. The orthosteric pockets are outlined by a white dotted line.

hP2X1 did not affect the apparent properties of activation, the molecular explanation for the difference in cooperativity of activation between hP2X1 and hP2X3 is unclear.

Despite the similarities in the extracellular domain and sequence conservation, the orthosteric pocket in hP2X1 is much smaller than that of hP2X3 in both the apo closed and ATP-bound desensitized states (Fig. 4, Supplementary Fig. 8). The apparent reductions in size of the orthosteric pocket in the ATP-bound desensitized state structures of hP2X1 compared to hP2X3 can likely be explained by different identities, residue insertion/deletions, and rearrangements of loops surrounding the orthosteric ATP-binding site. In hP2X1, the loop in the head domain, closest to the orthosteric pocket that contains K140, extends into the binding pocket to a larger extent than the same loop in hP2X3 (Fig. 4, Supplementary Fig. 3A). This movement is likely caused by the interaction between K140 and ATP as well as the insertion of two residues into this loop in hP2X1 that are not found in hP2X3 (Fig. 4C, D, Supplementary Fig. 8). A second loop, found on the most extracellular part of the right flipper, also extends into the orthosteric pocket of hP2X1 more than the same loop does in hP2X3, likely caused by a two residue insertion found in hP2X1 (Fig. 4C, D, Supplementary Fig. 3A, 8). Finally, the left flipper of hP2X1, a functionally important domain known to impact receptor gating, lacks three residues compared to the left flipper of hP2X3, resulting in changes to the identities of residues that interact with ATP (Supplementary Figs. 3A, 8)[26,31].

Divalent cations are known to impact the sensitivity of P2XRs to ATP[32–34]. In the ATP-bound desensitized state structure of hP2X1, there is significant density for a metal ion, which is coordinated by the sidechain of D170 (distances of 2.6 Å and 2.9 Å for oxygens on the carboxylic acid sidechain) and to the γ-phosphate of ATP (2.2 Å) (Supplementary Fig. 9A,B). Given the coordination, the known modulatory role of Mg²⁺ on P2X1 function, and the known interactions

between Mg²⁺ and ATP, we chose to model the ion as a Mg²⁺ and validated it using the CheckMyMetal online server, where Mg²⁺ appears to be a plausible choice[35,36]. Since the source of ATP in the desensitized state structure originates from endogenous ATP released upon cell lysis, the higher concentration of Mg-ATP compared to free ATP within the cell further supported the modeling of a Mg²⁺ ion[36]. Similar densities have been found in the ATP-bound open and desensitized state structures of hP2X3 and were modeled as Na⁺, Mg²⁺, or Ca²⁺ ions where each is coordinated by D158 or water molecules (Supplementary Fig. 9C–E)[16,34]. The sidechain of D170 in hP2X1, corresponding to D158 in hP2X3, is positioned differently between structures of the two subtypes (Supplementary Figs. 8, 9)[16,34]. However, the two residues appear to serve a similar function to coordinate ions in the orthosteric binding pocket. Mutation of D170 to alanine in hP2X1 does not significantly impact the apparent affinity of ATP, a similar result to mutation of D158 in hP2X3 (Supplementary Fig. 7B)[34].

### Two molecules of NF449 bind to a previously uncharacterized P2XR competitive ligand-binding site

The small-molecule NF449 (4,4′,4″,4‴-(carbonylbis(imino-5,1,3-benzenetriylbis(carbonylimino)))tetrakis-benzene-1,3-disulfonic acid) is a well-known, high-affinity P2X1-selective antagonist (Supplementary Fig. 10A)[24,37,38]. However, the location of the NF449 binding site and the mechanism for its high-affinity antagonism are both unknown. We report the cryo-EM structure of NF449 bound to hP2X1 at 2.9Å (Fig. 5, Supplementary Figs. 1, 2, 10, 11, Supplementary Table 1). In our hands, NF449 inhibits hP2X1 with an apparent inhibitory potency (IC₅₀) of 0.80 ± 0.02 nM as measured by TEVC experiments, consistent with previously published values (Fig. 6A, B)[24,37]. Importantly, the NF449-bound inhibited state structure was determined after adding the antagonist to an apo closed state conformation, ensuring the resulting

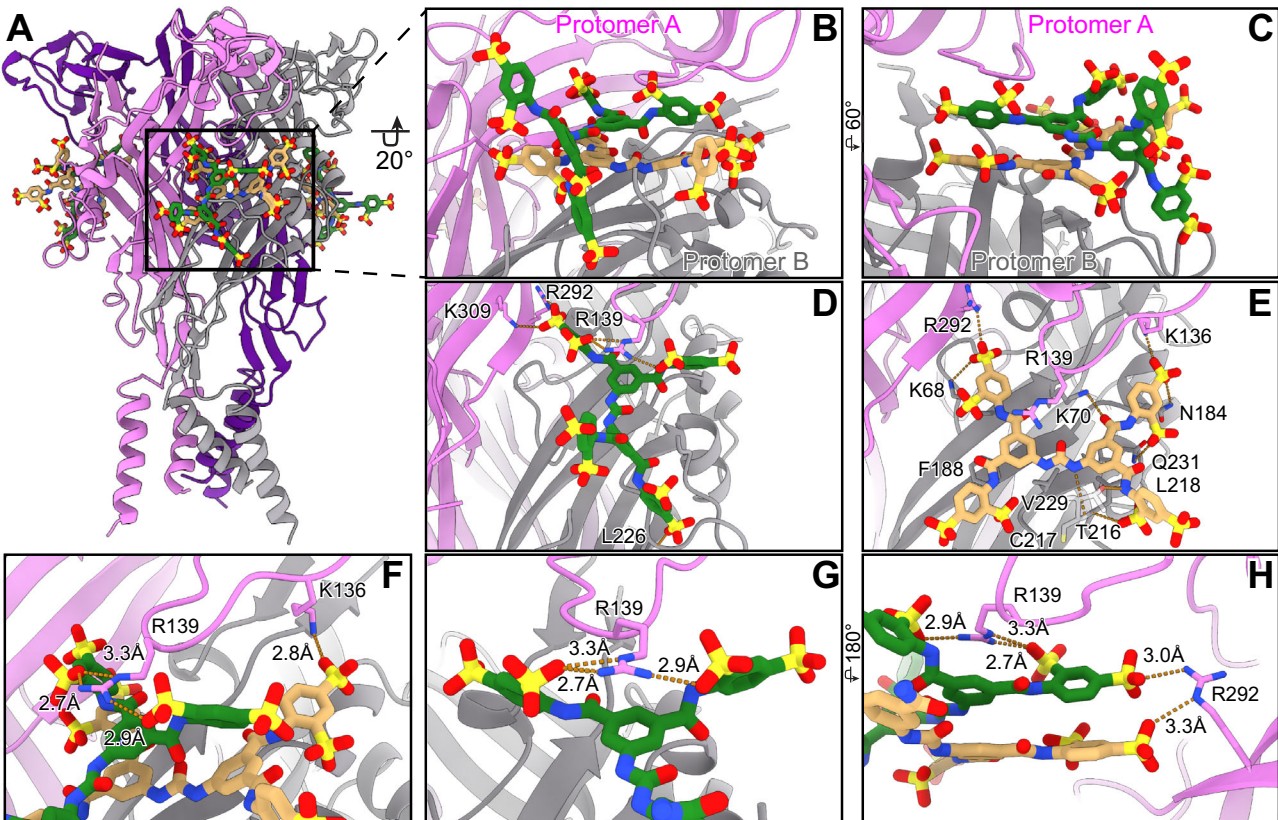

**Fig. 5 | NF449 inhibits hP2X1 in a supramolecular binding mode where the binding sites for two NF449 ligands overlap with the P2XR orthosteric ATP-binding site. A** Ribbon representation of NF449 bound to hP2X1 (shades of purple and gray), highlighting the location of two molecules of NF449 (colored tan and green) bound at one of three competitive ligand-binding sites. **B** Magnified and 20° rotated view of panel A highlighting the location and positioning of both NF449 molecules (inner NF449 in tan and outer NF449 in green) bound at one competitive ligand-binding site at the interface of two protomers (protomer A in pink and protomer B in gray). Importantly, moieties from both the inner and outer NF449 molecules bind within the orthosteric pocket. **C** View of the competitive ligand-binding site rotated 60° from panel B highlighting the stacked position of two NF449 molecules on top of each other. **D** A 45° rotated view of panel C,

highlighting the residues that interact with the outer NF449 molecule which include the sidechains of K309, R292, and R139, as well as the backbone nitrogen of L226. **E** A similar view from panel D, highlighting the residues that interact with the inner molecule of NF449 which include R292, K68, F188, R139, K136, N184, Q231, L216, T216, C217, V229, and K70. **F** Magnified view from panel E, highlighting the interactions between the sidechain of K136 and the inner molecule of NF449 as well as the sidechain of R139 and the outer molecule of NF449. **G** Magnified view from panel E, highlighting the critical interaction of the outer molecule of NF449 with the guanidinium group on the sidechain of R139. **H** A 180° rotated view from panel **G**, highlighting the interactions of the guanidinium group on the sidechain of R292 with both the inner and outer molecules of NF449.

antagonist-bound structure is not a mix of different conformational states (see Methods). Overall, the antagonist-bound structure is similar to the apo closed state structure (RMSD of 0.9 Å), except for notable rearrangements of several loops in the extracellular domain to accommodate NF449 binding. One such loop in the dorsal fin domain, residues 206-214, located directly over the NF449-binding site, has very weak density in the NF449-bound map and is not modeled due to its flexibility (Supplementary Fig. 3A). Attempting to model the loop into the unsharpened map confirms that the protein backbone appropriately spans the missing gap, as if to create a belt around the NF449-binding site (data not shown).

Critically, two molecules of NF449 bind at the interface between neighboring protomers, overlapping with the orthosteric ATP-binding site (Fig. 5A–C, Supplementary Figs. 10, 11). This competitive ligand-binding site ties together the dorsal fin, right flipper, upper body, and lower body domains from one protomer (referred to as protomer A) and the head, left flipper, and right flipper domains from the adjacent protomer (referred to as protomer B) (Fig. 5A, Supplementary Figs. 3A, 11A, B). The "inner" NF449 is bound closer to the receptor, where one phenyl disulfonate arm of the molecule binds into the orthosteric pocket, occupying the same space that ATP would otherwise occupy (Fig. 5B, C, E, Supplementary Fig. 11A, B). Thus, NF449 should be considered a competitive antagonist (Supplementary Fig. 11A, B). The

"outer" NF449 is positioned directly over the inner NF449, occupying the orthosteric site but extending further away from the receptor and into the extracellular space (Fig. 5B–D).

The inner NF449 molecule forms extensive contacts with hP2X1, resulting in a considerable number of interactions (Fig. 5E, Supplementary Fig. 10B). Most of the ligand-protein contact is formed on protomer B with some interactions formed with the adjacent protomer A (Fig. 5B, E). The phenyl disulfonate arm closest to the top of the receptor forms an ionic interaction with the ammonium group on the K136 sidechain of protomer A (2.8 Å), a hydrogen bonding interaction with the nitrogen on the sidechain amide of N184 (2.8 Å) as well as a polar-polar interaction with the carbonyl and a hydrogen bond interaction with the hydroxyl of its associated glycosyl group on protomer B (2.9 and 3.0 Å, respectively) (Fig. 5E, F). Additionally, this phenyl disulfonate arm forms polar-polar and hydrogen bonding interactions with the oxygen and nitrogen on the sidechain amide of Q231 on protomer B (2.8 Å and 3.4 Å) (Fig. 5E). Next, the carbonyl oxygen on the adjacent amide on this arm of NF449 forms a hydrogen bonding interaction with the ammonium group on the sidechain of K70 on protomer B (3.5 Å) (Fig. 5E). On the same side of the inner NF449, but on the opposing phenyl disulfonate arm, the ligand forms a hydrogen bonding interaction with the backbone amide nitrogen of L218 on protomer B (3.3 Å) (Fig. 5E). Further, the adjacent amide nitrogen on this arm forms a

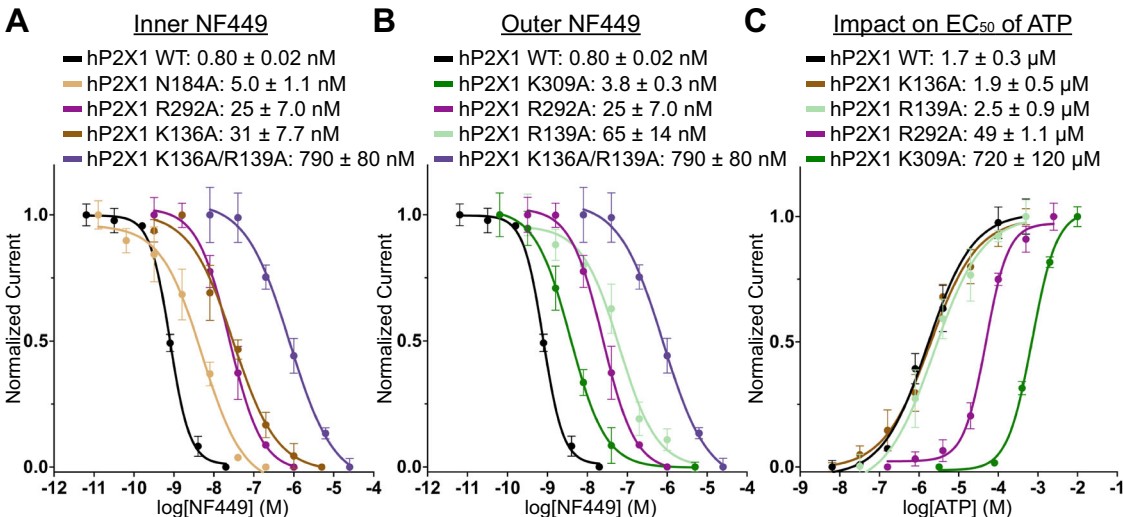

**Fig. 6 | Mutation of residues in hP2X1 that interact with the inner or outer NF449 molecules affect inhibitory potency (IC$_{50}$). A** Inhibition dose response curves for wild-type and mutant hP2X1 recorded by TEVC highlighting the impact on NF449 inhibition after mutation of specific residues shown to interact with the inner NF449 molecule. Each mutation (N184A, R292A, K136A, and K136A/R139A) targets a residue that interacts with the inner molecule of NF449 and results in the reduction of the inhibitory potency (IC$_{50}$) of NF449 at hP2X1 receptor. **B** Inhibition dose response curves for wild-type and mutant hP2X1 receptors recorded by TEVC highlighting the impact on NF449 inhibition after mutation of specific residues shown to interact with the outer NF449 molecule. Each mutation (K309A, R292A, R139A, and K136A/R139A) targets a residue that interacts with the outer molecule of NF449 and results in the reduction of inhibitory potency (IC$_{50}$) of NF449 at

hP2X1 receptor. **C** Dose-response curves for wild-type and mutant hP2X1 activated by ATP highlighting the impact of mutations, known to affect NF449 binding, on the activation of the receptor. Residues R292 and K309 are known to affect ATP binding and thus mutation of these residues reduces the apparent affinity (EC$_{50}$) for ATP activation. In contrast, residues K136 and R139 coordinate NF449 outside of the orthosteric pocket, and thus mutation of these residues does not impact the EC$_{50}$ of ATP to hP2X1. **A–C** The double mutation K136A/R139A abrogates key interactions to both molecules of NF449 and dramatically impacts NF449 inhibition more than either of the single mutations. Data points and error bars represent means and standard deviations of normalized current across triplicate experiments, respectively.

hydrogen bonding interaction with the sidechain hydroxyl of T216 of protomer B (2.5 Å) (Fig. 5E). The phenyl group in the center forms a π-stacking interaction with F188 on protomer B (4.8 Å) (Fig. 5E). Moving to the center of the inner NF449 ligand, the central urea group forms a hydrogen bonding interaction with the backbone carbonyl of C217 (2.8 Å) and a hydrophobic interaction to V229 on protomer B (Fig. 5E). The phenyl group on the other side of the urea group forms a hydrophobic interaction with L218 from protomer B (3.6 Å). Finally, one of the phenyl disulfonate arms on the opposite side of the molecule, that occupies the orthosteric pocket forms ionic interactions with the guanidinium group of R292 (3.3 Å) and the ammonium group on K68 (3.4 Å), both located on protomer A (Fig. 5E,H).

We performed TEVC studies with mutant hP2X1 constructs to validate residues predicted from the structure to be important for binding the inner NF449 molecule (Fig. 6). Residue N184 appears to play a minor role in coordinating high-affinity NF449 binding, as its mutation induced only a ~6-fold reduction in the inhibitory potency of NF449 (Fig. 6A). The R292A mutation was found to reduce both the inhibitory potency of NF449 and the apparent affinity of ATP by 31-fold and 32-fold, respectively (Fig. 6A, C). Since the mutation affects both the IC$_{50}$ of NF449 inhibition and the EC$_{50}$ of ATP activation, we cannot evaluate the contribution of R292 to high-affinity NF449 binding. Finally, the K136A mutation reduced the IC$_{50}$ by 39-fold but did not affect the EC$_{50}$ value for ATP, suggesting that it plays an important role in the coordination of NF449 binding (Fig. 6A, C).

The outer NF449 molecule is positioned further into the extracellular space and forms fewer interactions with hP2X1, as it is positioned above the inner NF449 (Fig. 5B–D, Supplementary Fig. 10C). Compared to the orientation of the inner NF449, the outer NF449 is rotated about its center by ~90° in the horizontal plane of the molecules (Fig. 5B). The phenyl disulfonate on the outer NF449 closest to the orthosteric pocket forms ionic bonds with the guanidinium group on the sidechain of R292 (3.1 Å) and the ammonium group on the

sidechain of K309 (3.3 Å), both located on protomer A (Fig. 5D, H). Mutations R292A and K309A impacted the ability of NF449 to inhibit hP2X1 activation by ATP, resulting in reductions to the inhibitory potency (IC$_{50}$) of the ligand by 31-fold and 5-fold, respectively (Fig. 6B). However, the same mutations also impacted the apparent affinity of ATP to the receptor, complicating comparisons (Fig. 6C). Critically, the sidechain guanidinium group of R139 from protomer A is inserted between the two opposing phenyl disulfonate groups on the same side of the outer NF449, where it forms two ionic interactions with the aforementioned arm (2.7 Å and 3.3 Å) and one with the opposing arm (2.9 Å) (Fig. 5E–H). The R139A mutation, which interacts only with the outer NF449, greatly reduced the inhibitory potency of NF449 (46-fold reduction) without affecting the EC$_{50}$ of ATP activation (Fig. 6B,C). The guanidinium group of R139 is also positioned directly over a phenyl ring of this NF449 molecule to form cation-pi interactions ( ~3.6 Å), but is too far from the inner molecule of NF449 to form any interactions (Fig. 5H). In addition to interactions with R139, the phenyl disulfonate arm on this side of the outer NF449 forms an interaction with the backbone carbonyl of K215 of protomer B (2.9 Å). The phenyl disulfonate arm on the opposite side of NF449, proximal to the membrane, forms a hydrogen bonding interaction with the backbone amide nitrogen of L226 on protomer B (3.2 Å). Finally, the double mutation K136A/R139A, which removes residues responsible for interacting with either the inner or the outer NF449 molecules, dramatically decreases the inhibitory potency by 990-fold, more than either of the single mutations alone (Fig. 6A, B).

In addition to NF449-hP2X1 interactions, there are numerous inter-ligand interactions between the two NF449 molecules (Fig. 5B, C). While both NF449 molecules interact with the receptor primarily via hydrogen bonding interactions, inter-ligand interactions consist of aromatic-aromatic interactions. In total, four phenyl groups per NF449 molecule are engaged in one set of π-π stacking interactions (3.9 Å) and three sets of edge-to-face interactions (4.6 Å, 4.7 Å, and 4.5 Å) between NF449

molecules. Additionally, there is one hydrogen bonding interaction between one of the sulfonate groups on the inner molecule to an amide nitrogen on the outer molecule of NF449 (2.8 Å). Altogether, NF449 leverages a unique dual-ligand binding mode with extensive receptor-ligand and ligand-ligand interactions, occupying a competitive ligand-binding site within hP2X1 that overlaps with the orthosteric ATP-binding site. This complex binding mode with two ligands occupying the same binding site is referred to as a supramolecular binding mode or a composite protein multiple ligand (COLIG) interaction[39,40].

The unique dual-ligand supramolecular binding mode may provide a structural basis for the subtype-specific, high-affinity antagonism of hP2X1 by NF449[24]. Compared to hP2X1, notable structural and compositional differences found in the apo closed states of hP2X3, zfP2X4, and rP2X7 likely diminish the specificity of NF449 for these P2XR subtypes (Supplementary Fig. 12)[16,29,41]. Residues K136 and R139 in hP2X1, important for ionic interactions with the inner and outer NF449, respectively, are not conserved in other P2XR subtypes (Supplementary Fig. 8)[26]. In P2X3, despite similar arrangements of secondary structures surrounding the competitive ligand-binding site, a moderate decrease in inhibitory potency is observed, likely a result of differences in residue identities (Supplementary Figs. 8, 12B)[24]. In the apo closed state of zfP2X4, an alpha helix in the dorsal fin domain overlaps with both binding sites of NF449 (Supplementary Figs. 3A, 12 C)[41]. Unsurprisingly, NF449 is six orders of magnitude less effective at inhibiting rat P2X4 than our measured $IC_{50}$ for hP2X1[24,42]. Finally, in rP2X7, additional bulky residues in the left flipper domain would be predicted to clash with the binding of both molecules of NF449, and correspondingly, there is a ~50-fold decrease in the apparent inhibitory potency of NF449 at human P2X7 relative to hP2X1 (Supplementary Figs. 3A, 12D)[37].

## Discussion

Our high-resolution apo closed, ATP-bound desensitized, and NF449-bound inhibited state structures of hP2X1 elucidate the molecular pharmacology of a previously uncharacterized P2XR subtype and provide critical details of the human ortholog necessary for structure-based drug design. We identified a generalizable method to prevent endogenous ligands from altering conformational states during purification. As a direct result, we isolated the apo closed state conformation of hP2X1, which in turn allowed us to produce and evaluate other conformational states. Thus, we identify subtype-specific residues in hP2X1 that define the closed pore in both the apo closed and ATP-bound desensitized state structures and residues that establish key interactions with ATP in the orthosteric pocket. Finally, the NF449-bound inhibited state structure of hP2X1 reveals a unique dual-ligand supramolecular binding mode at a previously uncharacterized competitive ligand-binding site that overlaps with the orthosteric ATP-binding site.

Structural studies of desensitizing P2XRs are complicated by the release of endogenous ATP during purification[16]. During cell lysis, significant amounts of intracellular ATP (millimolar concentrations) are released, leading to the activation and subsequent desensitization of P2XRs[43]. As a result, the majority of P2XRs are purified in the ATP-bound desensitized state conformation, complicating structural studies of apo closed and ligand-bound inhibited states. To overcome this obstacle, we incubated cells expressing hP2X1 receptor with NF449, a high-affinity competitive antagonist, prior to cell lysis and then subsequently removed it via extensive dialysis of harvested cell membranes. We confirmed that the resultant purified hP2X1 receptor was indeed in the apo closed state conformation, with the associated cryo-EM maps lacking density for ATP or NF449. To obtain the inhibitor-bound structure, NF449 was subsequently added in four-fold molar excess to the concentration of purified apo closed state receptor prior to vitrification. Generating ligand-bound structures from a true apo state conformation avoids structural artifacts from receptor desensitization and ensures all structural movements and ligand-interactions

to be more relevant. Altogether, the application of ligands prior to cell lysis and subsequent removal by dialysis can be a generalizable approach for preventing endogenous ligands from altering conformational states during purification.

While our structural studies were performed on the full-length wild-type hP2X1 receptor, residues in the N- and C-termini that comprise the cytoplasmic domain of the receptor were not visualized in either the apo closed or ATP-bound desensitized state structures (Figs. 1A, 2A, 7). A similar lack of density for cytoplasmic residues in the maps of the apo closed and ATP-bound desensitized state structures of hP2X3 was previously observed, suggesting the cytoplasmic residues are flexible or disordered in these conformational states for both hP2X1 and hP2X3 receptors[16]. In contrast, in the ATP-bound open state structure of hP2X3, the cytoplasmic residues were ordered (and thus visible in the maps) to form a domain called the 'cytoplasmic cap'[16]. We speculate that hP2X1 undergoes a gating process similar to hP2X3, where the cytoplasmic domain, shown previously to be a determinant of channel dynamics that sets the rate of receptor desensitization, is transiently assembled upon transition from the apo closed state to the ATP-bound open state and then disassembled upon transition to the ATP-bound desensitized state[16]. The short-lived lifetime of the cytoplasmic cap, therefore, is the reason that P2X1 and P2X3 receptors undergo fast rates of desensitization[16]. While our current structural evidence for hP2X1 is consistent with the so-called 'helical recoil model of receptor desensitization', further structural experiments to capture an ATP-bound open state structure of hP2X1 will provide more specific answers on the gating mechanisms for this P2XR subtype[16].

The apo closed state structure of hP2X1 reveals two CHS binding sites in the TMD on the extracellular leaflet of the membrane, likely representing cholesterol binding sites in vivo. From our purification process, we have observed that varying levels of cholesterol hemisuccinate significantly affected protein quality and stability. Moreover, our mutational data indicate that the many interactions between the receptor and the two CHS molecules provide stabilizing interactions within the TMDs (Fig. 1B, Supplementary Figs. 4, 5). We speculate these CHS binding sites are occupied by native cholesterol in vivo, where they serve a regulatory role in hP2X1 function. To this point, the depletion of cholesterol has been shown to negatively impact P2X1 receptor currents[44,45]. A putative role for cholesterol in the function of other P2XR subtypes has also been reported. For example, P2X7 receptor activation and signaling is sensitive to varying levels of cholesterol, where cholesterol even functions as a regulator of receptor pore formation[46]. Now that we have identified two putative cholesterol binding sites, more directed experiments can be performed to understand the regulatory role of cholesterol on P2X1 activity.

The presence of two NF449 molecules in one binding pocket highlights a dual-ligand supramolecular binding mode that underlies the structural basis for its high potency (Figs. 5, 7)[24,37]. Dissimilar to all current structures of antagonist-bound P2XRs where one ligand is bound per protomer, our structure surprisingly contains two molecules of NF449 bound to each protomer, which form extensive ligand-receptor and ligand-ligand interactions, for a total of six NF449 molecules bound per receptor[47–50]. During the initial modeling of the cryo-EM map of NF449-bound to hP2X1, we only placed the inner molecule of NF449 but found our structural and functional data to be more consistent with a dual-ligand supramolecular binding mode. For example, with only the inner NF449 in the model, unexplained density above the placed ligand cannot be adequately modeled by the path of the protein backbone (residues 206-214), and the density resembles the distinctive shape of phenyl disulfonate groups (Supplementary Fig. 11). Functionally, mutation of R139, a residue which does not interact with the inner NF449 molecule, and mutation of K136, a residue which does not interact with the outer NF449 molecule, both dramatically affect ligand potency without impacting the apparent affinity of ATP, results not consistent with a singly-bound NF449

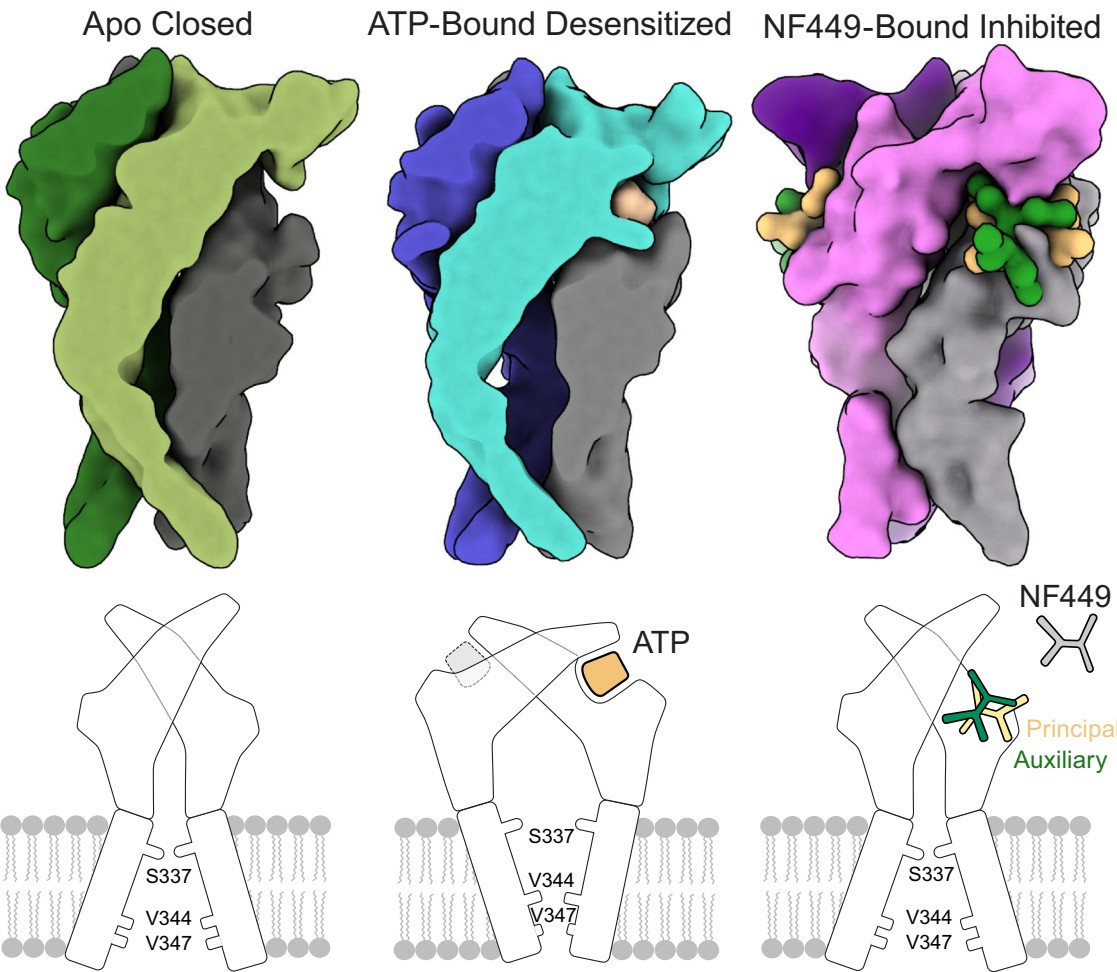

**Fig. 7 | Three distinct functional states of hP2X1 visualized by cryo-EM.** Schematic representation of the three identified functional states of hP2X1 (top) created using low resolution synthetic maps generated from the structural models and (bottom) represented as cartoon models. (**Left**) The apo closed state of hP2X1 is defined by empty ligand-binding sites and a closed gate, formed by S337 from each protomer. Due to the shallow gate in this conformational state, the overall architecture of the TMD resembles a "teepee". (**Middle**) The desensitized state of hP2X1 is defined by ATP bound to the orthosteric pocket and a closed gate, formed by

V344 from each protomer. Due to the deeper gate in this conformational state, the overall architecture of the TMD resembles a "cone". (**Right**) The NF449-bound inhibited structure of hP2X1 reveals a dual-ligand supramolecular binding mode where two molecules of NF449 (inner NF449 colored in tan and outer NF449 colored in green) interact and bind to hP2X1 at a pocket that overlaps with the orthosteric ATP-binding site. We define the inner NF449 molecule as the "principal" ligand and the outer NF449 molecule as the "auxiliary" ligand.

molecule when taken together with our structural data (Fig. 6). The double mutation of K136A/R139A affects the ligand potency of NF449 even more than each of the single mutations (Fig. 6). The importance of these two residues (as well as four additional residues) to NF449 binding have been previously identified in mutational studies, consistent with our structural and functional results[42,51]. Altogether, our structural and functional data are entirely consistent with a second NF449 being present in the binding pocket (Figs. 5–7).

Complex dual-binding ligands, while novel to P2XRs, have been previously described[39,40]. COLIGs and supramolecular binding modes are defined by multiple interacting ligands, often via extensive aromatic-aromatic interactions, that occupy the same binding pocket[39,40]. Since two NF449 ligands interact to occupy a single binding site in hP2X1, they leverage a supramolecular binding mode and can also be classified as a COLIG interaction[39,40]. To further clarify the role of each NF449 molecule, we define the inner NF449 molecule as the "principal" ligand, as it has greater overlap with the orthosteric ATP-binding site, maintaining more direct contacts with the receptor and acting as the primary competitive inhibitor. We define the outer NF449 molecule as the "auxiliary" ligand, where it likely stabilizes binding of the principal ligand and increases inhibitory potency (Fig. 7).

The unique dual-ligand supramolecule binding mode of NF449 provides the molecular basis for its sub-nanomolar antagonism specific to hP2X1. Since the principal NF449 molecule interacts with a large set of conserved residues in and around the orthosteric pocket, it is likely capable of binding different P2XR subtypes as a single distinct ligand, albeit with a lower potency. On the other hand, the auxiliary ligand is probably responsible for the dramatically higher affinity at hP2X1 because it significantly increases the number of contacts with the receptor and with the principal ligand. Together, the number of total ligand-receptor and ligand-ligand interactions help explain the high selectivity and inhibitory potency (sub-nanomolar IC$_{50}$) of NF449 for hP2X1. The structures of hP2X1 in three distinct conformational states provide insight into the unique characteristics and pharmacology of the human ortholog for this P2XR subtype, empowering drug discovery efforts to develop P2X1-specific small-molecule ligands.

## Methods
### Ethical statement
Unfertilized *Xenopus laevis* oocytes were purchased through Ecocyte Bioscience and kept at 18 °C until injection. This research complies with all relevant ethical regulations. All surgical procedures for

isolation of *Xenopus laevis* oocytes were done in accordance with animal welfare laws, followed national and institutional guidelines for humane animal treatment and complied with relevant legislation. Ecocyte Bioscience protocols help reduce the stress and harm on the laboratory animals, and appropriate aftercare such as pain management is employed to further minimize the impact of surgeries on the animals.

## Cell lines
SF9 cells were cultured in SF-900 III SFM (Fisher Scientific) at 27 °C. Baculovirus was expressed in cells of female origin. HEK293 GNTI⁻ cells were cultured using Gibco Freestyle 293 Expression Medium (Fisher Scientific) at 37 °C supplemented with 2% v/v fetal bovine serum (FBS). Receptor was expressed in HEK293 cells of female origin.

## Receptor constructs
The full-length wild-type hP2X1 receptor construct used for structure determination contains a C-terminal GFP, a 3C protease site, and a histidine affinity tag for purification. No mutations or truncations were made to the receptor for structure determination. For electrophysiology experiments, the hP2X1-WT construct is not genetically mutated − it is full-length wild-type hP2X1 receptor with no GFP, protease sites, or affinity tags present. Mutational data for electrophysiological experiments were performed using QuikChange XL mutagenesis kits (Agilent Technologies) from this wild-type construct. Constructs with mutations included: hP2X1-I45A, hP2X1-F49A, hP2X1-L50A, hP2X1-Y55A, hP2X1-K136A, hP2X1-R139A, hP2X1-K140A, hP2X1-D170A, hP2X1-N184A, hP2X1-K249A, hP2X1-R292A, hP2X1-K309A, and hP2X1-K136A/R139A.

## Receptor expression and purification
The full-length wild-type hP2X1 receptor construct was expressed by baculovirus mediated gene transfection (BacMam). Briefly, HEK293 GNTI⁻ cells were grown in suspension to a sufficient density and infected with P2 BacMam virus. Following overnight growth at 37 °C, sodium butyrate was added to a final concentration of 10 mM, and cells were shifted to 30 °C for an additional 48 h. After expression, different methods were used to acquire hP2X1 in each conformational state as described below.

For the ATP-bound desensitized state structure, cells expressing hP2X1 were harvested by centrifugation, washed with PBS buffer (137 mM NaCl, 2.7 mM KCl, 8 mM $Na_2HPO_4$, 2 mM $KH_2PO_4$), suspended in TBS (50 mM Tris pH 8.0, 150 mM NaCl) containing protease inhibitors (1 mM PMSF, 0.05 mg/mL aprotinin, 2 mg/mL Pepstatin A, 2 mg/mL leupeptin), and broken via sonication. Intact cells and cellular debris were removed by centrifugation and then membranes isolated by ultracentrifugation. The membranes were snap frozen and stored at −80 °C until use. When ready, membranes were thawed, resuspended in TBS buffer containing 15% glycerol, dounce homogenized, and then solubilized in 40 mM dodecyl-β-D-maltopyranoside (DDM or C12M) and 8 mM cholesterol hemisuccinate tris salt (CHS). The soluble fraction was isolated by ultracentrifugation and incubated with TALON resin in the presence of 10 mM imidazole at 4 °C for 1-2 hours. After packing into an XK-16 column, the purification column was washed with 2 column volumes of buffer (TBS plus 5% glycerol, 1 mM C12M, 0.2 mM CHS at pH 8.0) containing 20 mM imidazole, 10 column volumes containing 30 mM imidazole, and eluted with buffer containing 250 mM imidazole. Peak fractions containing the protein were concentrated and digested with HRV 3C protease (1:25, w/w) at 4 °C overnight. The digested protein was then ultracentrifuged and injected onto a Superdex 200 increase 10/300 GL column for size exclusion chromatography (SEC), equilibrated with 20 mM HEPES, pH 7.0, 100 mM NaCl, and 0.5 mM C12M with 0.1 mM CHS. Fractions were analyzed by SDS-PAGE and fluorescence size exclusion chromatography (FSEC), pooled accordingly, and concentrated for cryo-EM grid preparation.

To achieve the apo closed and NF449-bound inhibited state structures, cells expressing hP2X1 were grown and harvested, as described above. However, the high-affinity antagonist NF449 was incubated with the harvested cells at a concentration of 4.3 μM for 1 hour prior to lysis via sonication. Intact cells and cellular debris were removed by centrifugation and then membranes isolated by ultracentrifugation. The membranes were snap frozen and stored at −80 °C until use. When ready, the NF449-bound hP2X1 membranes were thawed, dounce homogenized, and then dialyzed for seven days in 10 kD MWCO tubing in 5 L of buffer (1 M NaCl, 150 mM Tris, 5% glycerol, pH 9.5) changing the buffer twice a day. For the last day of dialysis, the buffer was restored to pH 8 before proceeding with purification[16]. The subsequent purification steps were identical to the endogenous ATP-bound purification, as detailed above.

## Electron microscopy sample preparation
For the NF449-bound sample, 400 μM of the antagonist was incubated with hP2X1 receptor in the apo closed state conformation for 1 hour. This represents ~4x molar excess NF449 over the concentration of hP2X1 protomer. For the apo and ATP-bound cryo-EM samples, no ligand was added. All samples were prepared at 5 mg/mL and subjected to ultracentrifugation (62,000 x *g*, 1 hour) prior to vitrification. For all samples, 2.5 μL of sample was applied to a glow-discharged (15 mA, 1 min) Quantifoil R1.2/1.3 300 mesh gold holey carbon grid which was blotted for 1.5 s under 100% humidity at 6 °C. The grids were flash frozen in liquid ethane using a FEI Vitrobot Mark IV and stored under liquid nitrogen until screening and large-scale data acquisition.

## Electron microscopy data acquisition
Cryo-EM datasets for all receptor complexes were collected on Titan Krios microscopes (FEI) operated at 300 kV at the Pacific Northwest Center for Cryo-EM (PNCC). Datasets were acquired on microscopes with an energy filter (Gatan Image Filter) at 10 eV (apo closed state) or 20 eV (ATP-bound desensitized and NF449-bound inhibited states) slit width and a Gatan K3 direct-electron detector. Movies were collected in super-resolution mode (ATP-bound dataset) or at the physical pixel size (apo and NF449-bound datasets) at a nominal magnification of 130,000x, corresponding to a physical pixel size of ~0.648 Å/pixel, using a defocus range of −0.8 to −1.5 μm, 48 (ATP-bound dataset) or 50 frames (apo closed and NF449-bound inhibited datasets), and total dose of ~45 e⁻/Å². Each dataset utilized 'multi-shot' and 'multi-hole' collection schemes driven by serialEM to maximize high-throughput data collection[52].

## Electron microscopy data processing
After data collection, movies were motion corrected in cryoSPARC V4 using patch motion correction and micrographs generated at the physical pixel size (Supplementary Fig. 1, Supplementary Table 1)[53]. The contrast transfer function (CTF) parameters were estimated in cryoSPARC and micrographs were manually curated. Particles were then picked using 2D-templates, curated, extracted, and sent directly to 3D-classification (skipping 2D-classification). The classification procedure involved using iterative ab initio and heterogenous classifications to remove bad particles (Supplementary Fig. 1). The final homogenous particle stacks were re-extracted at the physical pixel size (0.648 Å/pixel), CTF correction at the global and local scales performed, and a final non-uniform refinement generated the consensus cryo-EM map (Supplementary Figs. 1, 2, Supplementary Table 1). Some maps were sharpened using DeepEMhancer[54].

## Model building and structure determination
Homology models generated from SWISS-MODEL using the apo closed or ATP-bound desensitized state structures of hP2X3 were built in Coot

v0.9.8 (PDB codes: 5SVJ and 5SVL, respectively)[55,56]. The CIF file for the NF449-bound structure was built in eLBOW with protonation states corresponding to approximately pH 7[57]. All stages of model building involved manual adjustments based on the quality of the maps in Coot, followed by real space refinement in PHENIX v1.18[58]. Limited glycosylations were included in the models when justified by density. In some of the models, residues or their sidechains are not included if they were missing from the density. Model quality was evaluated by MolProbity (Supplementary Table 1)[59].

## Two-electrode voltage clamping

**Preparation of oocytes expressing hP2X1.** Oocytes were purchased defolliculated from Ecocyte Bioscience in Barth's Solution (88 mM NaCl, 1 mM KCl, 0.82 mM $MgSO_4$, 0.33 mM $Ca(NO_3)_2$, 0.41 mM $CaCl_2$, 2.4 mM $NaHCO_3$, 5 mM Tris-HCl) supplemented with penicillin (100 U/mL) and streptomycin (100 μg/mL). *Xenopus laevis* oocytes were then injected with 50 nL of 100 ng/μL hP2X1 mRNA that was made from linearized full-length wild-type or mutant pcDNA 3.1x according to the protocol provided in the mMessage mMachine kit (Invitrogen). Injected oocytes were allowed to express for ~20 hours at 18 °C before recording was performed.

**TEVC Recordings.** Data acquisition was performed using the Oocyte Clamp OC-725C amplifier and pClamp 8.2 software. Buffers were applied using a gravity fed RSC-200 Rapid Solution Changer that flows at ~5 mL/min. All experiments use Sutter filamented glass 10 cm in length with an inner diameter of 0.69 mm and an outer diameter of 1.2 mm to impale oocytes and clamp the holding voltage at −60 mV. Experiments were recorded in buffer containing 10 mM HEPES (pH 7.4), 140 mM NaCl, 5 mM KCl, 2 mM $CaCl_2$, 2 mM $MgCl_2$, and 10 mM glucose.

**Dose response ($EC_{50}$) experiments.** Excitatory responses to a dilution series of ATP between 6.4 nM and 10 mM were examined across the different mutants. Each evoked response was normalized to an initial ATP-induced current evoked by the following concentrations: 500 μM ATP for R292A, 2 mM ATP for K309A, and 100 μM ATP for the remaining mutants and wild-type receptor. The data was fit in Prism 9 using the nonlinear regression named "$EC_{50}$, x is concentration" to afford $EC_{50}$ values. This value was then averaged amongst each singular condition and reported as an average plus or minus the standard deviation.

**Inhibition dose response ($IC_{50}$) experiments.** To test a dilution series of an antagonist at discrete concentrations, 10 μM ATP (2 mM ATP for K309A only) was first applied to evoke an initial maximum current. The oocyte was then washed with buffer for one minute, followed by one minute application of an antagonist test-concentration, and concluded by co-application of the same test concentration with 10 μM ATP (2 mM ATP for K309A). The antagonized signal was then normalized against the initial ATP excitatory signal. Each test concentration was performed in triplicate across different oocytes. The data was then fit to a nonlinear regression named "[inhibitor] vs. response – Variable slope (four parameters)" in Prism 9 to produce a sigmoidal curve and afford an $IC_{50}$ value. Each singular condition was used to generate an average which is reported plus or minus its standard deviation between three discrete trials.

**Maximal current experiments for CHS-coordinating mutations.** The effect of CHS-coordinating mutations (I45A, F49A, L50A, and Y55A) on hP2X1 receptor function was evaluated by TEVC. Mutant and wild-type mRNA (50 nL of 100 ng/μL) were injected into oocytes and incubated at 18 °C for 48 hours to allow sufficient receptor expression. Oocytes were initially potentiated by 10 μM ATP and followed by 2 min of buffer washing. The receptors were then activated again using 10 μM ATP and

the current amplitude was recorded ($n = 6$–10 for each receptor type). The magnitudes of the ATP-induced currents of hP2X1-WT and CHS-mutant constructs were statistically analyzed by performing a one-way Dunnett T3 multiple comparisons ANOVA test in Prism 9. A $p$ value less than 0.05 between groups was considered statistically significant.

## Reporting summary

Further information on research design is available in the Nature Portfolio Reporting Summary linked to this article.

## Data availability

The data that support this study are available from the corresponding author upon request. All cryo-EM density maps for the full-length wild-type hP2X1 receptor in the apo closed, ATP-bound desensitized, and NF449-bound inhibited states have been deposited in the Electron Microscopy Data Bank under the EMDB accession codes: EMD-45152 (apo closed state), EMD-45153 (ATP-bound desensitized state), and EMD-45154 (NF449-bound inhibited state). The maps within these depositions include both half maps, sharpened/unsharpened maps, refinement masks, and locally sharpened maps that helped with model building. The corresponding coordinates for the structures have been deposited in Protein Data Bank under the PDB accession codes: 9C2A (apo closed state), 9C2B (ATP-bound desensitized state), and 9C2C (NF449-bound inhibited state). The apo closed, ATP/Na$^+$-bound desensitized, ATP/Mg$^{2+}$-bound open, ATP/Ca$^{2+}$-bound open state structures of the hP2X3 receptor were obtained using the PDB codes 5SVJ, 5SVL, 6AH4, and 6AH5, respectively. The source data underlying Figs. 2B and 6 as well as Supplementary Figs. 5 and 6 are provided as a Source Data file. Source data are provided with this paper.

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

## Acknowledgements

We thank O. Davulcu, C. Yoshioka, and C. López at PNCC for access and microscopy assistance; A. Glasfeld for figure and manuscript feedback. Electron microscopy grid screening was performed at the Multiscale Microscopy Core within Oregon Health & Science University (OHSU). A portion of this research was supported by NIH grant U24GM129547 and performed at the PNCC at OHSU and accessed through EMSL (grid.436923.9), a DOE Office of Science User Facility sponsored by the Office of Biological and Environmental Research. This research was supported by the National Heart, Lung and Blood Institute (R00HL138129, S.E.M.), the National Institute of General Medical Sciences (DP2GM149551, S.E.M.), and the American Heart Association (24PRE1195450, A.C.O.).

## Author contributions

A.C.O. and S.E.M. designed the project. A.C.O., N.E.L., and H.S. performed sample preparation for cryo-EM studies. A.C.O. performed the cryo-EM data collection. A.C.O., N.E.L., and H.S. analyzed the cryo-EM data. A.C.O., N.E.L., and H.S. built the cryo-EM models. N.E.L., I.A.D., and N.A.N. performed molecular biology, mutagenesis, and made RNA for electrophysiology experiments. A.C.O., I.A.D., and N.A.N. performed and analyzed the electrophysiological experiments. All authors wrote and edited the manuscript.

## Competing interests

The authors declare no competing interests.
