## [Peer Review File · Nature Communications]

Cryo-EM structures of the human P2X1 receptor reveal subtype-specific architecture and antagonism by supramolecular ligand-bindingREVIEWER COMMENTS

Reviewer #1 (Remarks to the Author):

The paper by the Monsoor group showcases the first cryo-EM structures of the human P2X1 receptor in multiple conformations, including apo-closed, desensitized, and antagonist-bound closed states. These structures complement the currently available structures of the P2X3, P2X4, and P2X7 receptors, greatly enhancing our understanding of subtype-specific mechanisms. Detailed comparisons with other P2X subtypes reveal intriguing similarities and several unforeseen P2X1-specific features. Furthermore, the study discovered that the P2X1-specific drug, NF499, binds to the orthosteric site as a dimer rather than a monomer. The structures are determined at high resolutions, and the provided cryo-EM densities support the mode of drug binding. Overall, these structures offer excellent platforms for dissecting P2X1-specific mechanisms, potentially leading to the discovery of new drugs for treating diseases such as thromboembolism.

One major limitation of the current study is that all new insights are obtained based solely on structural investigations. The lack of functional studies render the potential P2X1-specific mechanisms untested speculations. Another limitation is the absence of an open state, which, based on the previous P2X3 structure, would include ordered cytoplasmic N- and C-termini. Since these termini play major roles in P2X1-specific characteristics such as fast desensitization, the manuscript would have been much stronger if it included this conformation. While these limitations do not necessarily diminish the significance of the presented structures, deeper mechanistic insights would be obtained if additional experiments were performed. Here are my specific comments and suggestions.

1. Clear densities corresponding to cholesteryl hemisuccinate (CHS) molecules are observed near the extracellular side of the membrane (e.g., Fig. 1). As mentioned in the discussion, it is possible that cholesterol molecules may bind to P2X1 at these positions to regulate channel activity. However, the cited previous studies by the Evans group suggest that residues near the intracellular side of the membrane (residues 20-23 and 27-29) play pivotal roles. Indeed, mutating these regions diminishes the inhibitory effect of cholesterol. It is possible that cholesterol regulates P2X1 in multiple different ways. Since the CHS interaction sites are clearly visible, testing a few mutants may uncover the significance of cholesterol binding at the extracellular side.

2. Since P2X1 and P2X3 exhibit similar channel properties, the stark difference in the transmembrane configurations of the apo-closed state between these two subtypes is somewhat surprising. Is this difference due to the interaction between P2X1 and CHS, which obviously stabilizes this protein? Again, testing several mutations of the CHS-coordinating residues may address this point.

3. Page 8, lines 255-262: The P2X1-specific residues in the ATP-binding pocket are introduced in this paragraph to suggest they play important roles in the low EC50. However, both P2X1 and P2X3 are reported to have very similar EC50s of 0.5-1.0 μM (e.g., doi: 10.1111/bph.15299). If EC50s are similar, it is possible that there is a mismatch between binding affinity and gating efficiency. It would be interesting to mutate these residues, making P2X1 more P2X3-like or vice versa, to determine EC50s. Additionally, a chemically related analog, α , β -MeATP, has been shown to be 10 times more potent for P2X1 than P2X3. Testing the P2X1-specific residues for this agonist may strengthen the claim that these residues play important roles in ligand recognition.

4. Page 9, line 282~: Divalent cation-bound P2X3 structures have been determined (doi.org/10.7554/eLife.47060). Interestingly, D158 in P2X3, which corresponds to D170 in P2X1, also coordinates magnesium ions. This section will improve if the authors include comments on the similarity between these two subtypes.

5. The supramolecular binding mode of NF499 is interesting. Previous studies by the Evans group suggested little cooperativity in ATP-dependent channel activity, but a slightly positive cooperativity

(Hill coefficient of ~ 1.4) for NF499 (doi.org/10.1074/jbc.M114.592246). It is possible that two molecules of NF499 bind to the same orthosteric site rather than in two different sites. The same study predicted similar binding modes in their docking study based on electrophysiological experiments. They also determined the six important binding residues confirmed by the current study. It would be appropriate to cite their study and include a discussion. Although it may be beyond the scope of the current study, it would be insightful to test the supramolecular binding mode of NF499 using concatemeric P2X1 channels that have mixed orthosteric sites including wild-type and the one in which the key NF499 binding residues are substituted with the P2X3 counterparts.

6. Page 6: While Figure 2 is cited in the correct order in the first paragraph of the Results section, it is awkward to read about Figure 3 before Figure 2 is introduced. I suggest swapping the figure order.

Reviewer #2 (Remarks to the Author):

In this work the authors have determined 3 cryoEM structures of human P2X1; an apo-state (with an elegant method employed to isolate this state), an ATP-bound structure consistent with the desensitised state, and a unique dual-ligand NF449-bound structure. The high resolution of the structures has permitted detailed observation and analysis of the ligand binding pockets, and probably the most surprising result is the finding of 2 NF449 molecules bound to distinct sites ('inner' and 'outer'). Although the authors have employed mutagenesis of one residue to show a reduction in antagonist potency, in my opinion this finding needs more supporting evidence (confirmation of the importance of the 'inner' binding mode by mutagenesis, and/or discussion/analysis of previous mutations in P2X receptors (which mutations might be studied without compromising ATP potency and channel function)).

In my opinion this a very well presented, important and interesting study that provides 3 novel 3D-structures of P2X1 and has given a detailed molecular understanding of how it can bind ATP and NF449. My specific comments are listed below.

Major comments

1. Results, line 312. The NF449 structure with two NF449 molecules bound is surprising, and while the authors present evidence that mutation of one amino acid residue observed to interact with the 'outer' NF449 (R139A) reduces NF449 potency, there are no further mutagenesis studies to confirm that the 'inner' binding mode is also important for NF449 potency (e.g. K136, N184, Q231, K70, T216, F188, V229, L218, R292, K68). Have the authors considered making some of these mutations (I am aware that some are necessary for the action of ATP, so would render the channel non-functional, but is that the case for all of them)?
2. Results, line 312. Is there any effect on ATP potency of the R139A mutation? If this mutation affects ATP potency, it will complicate the interpretation of the NF449 IC50 data.
3. Results, line 312. Could the dual-NF449 bound structure be an artefact of incubation with a high concentration of antagonist for structure determination?

Minor comments

4. Nomenclature. The modern naming convention for P2X receptors is not to use a subscript for the receptor subtype number (should be P2X1) and this should be followed throughout.
5. Introduction, first two paragraphs. It would be more succinct to briefly state the physiological roles of P2X1 and why it is a potential therapeutic target, and that a molecular understanding of antagonism is lacking, as the work presented does not contain any experimental data relating to platelet biology.
6. Introduction, paragraph 3. It may be factually accurate to state that 'it is not known ... why P2X7 receptor is the least sensitive to ATP' but it has been suggested that P2X7 can only bind ATP⁴⁻, and that there are differences in the structure surrounding the ATP binding site that can account for the relatively low sensitivity to ATP in this subtype.
7. Introduction, line 100. This should state 'will facilitate' rather than 'facilitates'.
8. Results, line 143. It would be better to move the statement in lines 164-167 to the start of this section.

9. Results, line 193. How do you know that the structure you observe is the desensitised state? Is it by analogy to the hP2X3 structure, or is it an assumption because P2X1 desensitises very quickly with ATP bound? I do not dispute that the authors are observing the desensitised state, but I think they should qualify this at the start of the section describing the structure.

10. Results, lines 282-291. Do the authors have any direct evidence that it is Mg-ATP rather than Na-ATP bound to the receptor? Can they elaborate on the CheckMyMetal results? I agree that it is more likely that it is Mg-ATP, but the authors should be clear about what is direct evidence and what is assumption.

Response to Reviews: Manuscript ID# NCOMMS-24-32951A

REVIEWER COMMENTS

Reviewer #1:

Overall:

(1) The paper by the Monsoor group showcases the first cryo-EM structures of the human P2X1 receptor in multiple conformations, including apo-closed, desensitized, and antagonist-bound closed states. These structures complement the currently available structures of the P2X3, P2X4, and P2X7 receptors, greatly enhancing our understanding of subtype-specific mechanisms. Detailed comparisons with other P2X subtypes reveal intriguing similarities and several unforeseen P2X1-specific features. Furthermore, the study discovered that the P2X1-specific drug, NF499, binds to the orthosteric site as a dimer rather than a monomer. The structures are determined at high resolutions, and the provided cryo-EM densities support the mode of drug binding. Overall, these structures offer excellent platforms for dissecting P2X1-specific mechanisms, potentially leading to the discovery of new drugs for treating diseases such as thromboembolism.

Response: We thank the reviewer for the nice comments and agree that these structures will provide the platform for dissecting P2X1-specific mechanisms and P2X1-specific ligands.

(2) One major limitation of the current study is that all new insights are obtained based solely on structural investigations. The lack of functional studies render the potential P2X1-specific mechanisms untested speculations. Another limitation is the absence of an open state, which, based on the previous P2X3 structure, would include ordered cytoplasmic N- and C-termini. Since these termini play major roles in P2X1-specific characteristics such as fast desensitization, the manuscript would have been much stronger if it included this conformation. While these limitations do not necessarily diminish the significance of the presented structures, deeper mechanistic insights would be obtained if additional experiments were performed. Here are my specific comments and suggestions.

Response: We agree that the functional data was rather limited in the original version of the manuscript. However, we are confident that the reviewer will agree we now include an abundance of functional data that corroborates our structural findings (discussed in more detail below).

Specific Comments:

(1) Clear densities corresponding to cholesteryl hemisuccinate (CHS) molecules are observed near the extracellular side of the membrane (e.g., Fig. 1). As mentioned in the discussion, it is possible that cholesterol molecules may bind to P2X1 at these positions to regulate channel activity. However, the cited previous studies by the Evans group suggest that residues near the intracellular side of the membrane (residues 20-23 and 27-29) play pivotal roles. Indeed, mutating these regions diminishes the inhibitory effect of cholesterol. It is possible that cholesterol regulates P2X1 in multiple different ways. Since the CHS interaction sites are clearly

visible, testing a few mutants may uncover the significance of cholesterol binding at the extracellular side.

Response: These are excellent suggestions. Most of the data in the literature suggests that there are cholesterol binding sites in the intracellular domain of P2X receptors. To our knowledge, our data presented here is the first to identify cholesterol binding sites near the extracellular surface of the transmembrane domains. We made mutations to several of the residues from our structure that form interactions with the CHS molecules and show the mutant hP2X1 receptors have significantly decreased ATP-induced activity relative to WT hP2X1 receptor. This is now discussed in lines 158-162 in the Results and lines 477-479 in the Discussion, and the data is shown in Supplementary Fig. 5. This finding now provides a basis for which additional studies can be designed to elucidate the molecular mechanism underlying the role of these cholesterol binding sites in modulating hP2X1 activity.

(2) Since P2X1 and P2X3 exhibit similar channel properties, the stark difference in the transmembrane configurations of the apo-closed state between these two subtypes is somewhat surprising. Is this difference due to the interaction between P2X1 and CHS, which obviously stabilizes this protein? Again, testing several mutations of the CHS-coordinating residues may address this point.

Response: We thank the reviewer for these excellent questions. We made mutations to CHS-interacting residues in the TMD to evaluate their importance to P2X1 function (lines 158-162 and Supplementary Fig. 5). As mentioned above, these residues appear to be involved in P2X1 function. However, we think the best way to evaluate the role of CHS in P2X3 is to solve the structure of full-length human P2X3 receptor using Cryo-EM under similar conditions. We feel future work on the potential role of cholesterol in P2X1 and possibly P2X3 merits its own investigation.

(3) Page 8, lines 255-262: The P2X1-specific residues in the ATP-binding pocket are introduced in this paragraph to suggest they play important roles in the low EC₅₀. However, both P2X1 and P2X3 are reported to have very similar EC₅₀s of 0.5-1.0 μ M (e.g., doi: 10.1111/bph.15299). If EC₅₀s are similar, it is possible that there is a mismatch between binding affinity and gating efficiency. It would be interesting to mutate these residues, making P2X1 more P2X3-like or vice versa, to determine EC₅₀s. Additionally, a chemically related analog, α , β -MeATP, has been shown to be 10 times more potent for P2X1 than P2X3. Testing the P2X1-specific residues for this agonist may strengthen the claim that these residues play important roles in ligand recognition.

Response: We thank the reviewer for these questions. We re-measured the EC₅₀ of ATP at hP2X1 and hP2X3 in the same buffer system and obtained very similar values for each (1.7 μ M for hP2X1 and 0.8 μ M for hP2X3) seen in Supplementary Fig. 7A, consistent with published values in the literature (lines 238-241). Additionally, we made mutations to the residues unique to hP2X1 receptor (K140A, D170A, and K249A) and saw no major effect on the EC₅₀ for ATP (Supplementary Fig. 7B, lines 254-256, 259-263, and 301-303). We think that insertions/deletions of loops for hP2X1 likely play a

key role in the difference between hP2X3 and alluded to this in the manuscript (lines 270-285).

(4) Page 9, line 282~: Divalent cation-bound P2X3 structures have been determined (doi.org/10.7554/eLife.47060). Interestingly, D158 in P2X3, which corresponds to D170 in P2X1, also coordinates magnesium ions. This section will improve if the authors include comments on the similarity between these two subtypes.

Response: We thank the reviewer for these comments and for identifying the manuscript by the Swartz group. We have now cited this manuscript and added a detailed analysis comparing the ion coordination for D170 in hP2X3 to D158 in hP2X3, including Supplementary Fig. 9 and lines 287-303.

(5) The supramolecular binding mode of NF499 is interesting. Previous studies by the Evans group suggested little cooperativity in ATP-dependent channel activity, but a slightly positive cooperativity (Hill coefficient of ~1.4) for NF499 (doi.org/10.1074/jbc.M114.592246). It is possible that two molecules of NF499 bind to the same orthosteric site rather than in two different sites. The same study predicted similar binding modes in their docking study based on electrophysiological experiments. They also determined the six important binding residues confirmed by the current study. It would be appropriate to cite their study and include a discussion. Although it may be beyond the scope of the current study, it would be insightful to test the supramolecular binding mode of NF499 using concatemeric P2X1 channels that have mixed orthosteric sites including wild-type and the one in which the key NF499 binding residues are substituted with the P2X3 counterparts.

Response: We agree that supramolecular binding of NF449 is interesting. Thank you for pointing out the excellent work by the Evans group, which we now cite and discuss (lines 504-506). Their paper did note several important residues for NF449 binding, which our structural and mutational studies are consistent with. Their docking studies, which showed a single NF449 interacting with most of these residues, broadly agree with our structure in that the inner NF449 has one arm in the orthosteric pocket and the rest of the molecule binds outside of the pocket. They likely did not consider the possibility of a second molecule in their modeling (as this is a surprising result).

We agree that it would be insightful to test the supramolecular binding mode using concatemeric P2X1. This is an excellent idea for an aim of a grant (thank you!), but we agree that it is not within the scope of the current work.

(6) Page 6: While Figure 2 is cited in the correct order in the first paragraph of the Results section, it is awkward to read about Figure 3 before Figure 2 is introduced. I suggest swapping the figure order.

Response: Thank you for the suggestion but we would prefer to keep the order as it is. If we swap Figures 2 and 3 then a figure comparing the desensitized states of hP2X1 and hP2X3 (panels D-F in the current Figure 3) will be introduced before our structure of

hP2X1 in the desensitized state is introduced. We think this would be even more awkward.

However, we could break the current Figure 3 into two separate figures, making panels A-C the new Figure 2 to come after the apo state structure of hP2X1 and panels D-F the new Figure 4 to come after the desensitized state structure of hP2X1. We are happy to do this if the reviewer feels strongly that it would help the flow of the manuscript (from a reader's perspective), but it would break up the simplicity and juxtaposition of having one figure that compares two different states of hP2X1 to hP2X3.

Response to Reviews: Manuscript ID# NCOMMS-24-32951A

REVIEWER COMMENTS

Reviewer #2:

Overall:

In this work the authors have determined 3 cryoEM structures of human P2X1; an apo-state (with an elegant method employed to isolate this state), an ATP-bound structure consistent with the desensitised state, and a unique dual-ligand NF449-bound structure. The high resolution of the structures has permitted detailed observation and analysis of the ligand binding pockets, and probably the most surprising result is the finding of 2 NF449 molecules bound to distinct sites ('inner' and 'outer'). Although the authors have employed mutagenesis of one residue to show a reduction in antagonist potency, in my opinion this finding needs more supporting evidence (confirmation of the importance of the 'inner' binding mode by mutagenesis, and/or discussion/analysis of previous mutations in P2X receptors (which mutations might be studied without compromising ATP potency and channel function)).

In my opinion this a very well presented, important and interesting study that provides 3 novel 3D-structures of P2X1 and has given a detailed molecular understanding of how it can bind ATP and NF449. My specific comments are listed below.

Response: We thank the reviewer for the very kind comments and for recognizing the method we employed to capture the true apo state. We agree that we needed more functional data to support our structural data. We are confident that the reviewer will agree we now include an abundance of functional data that corroborates our structural findings (discussed in more detail below).

Major comments

(1) Results, line 312. The NF449 structure with two NF449 molecules bound is surprising, and while the authors present evidence that mutation of one amino acid residue observed to interact with the 'outer' NF449 (R139A) reduces NF449 potency, there are no further mutagenesis studies to confirm that the 'inner' binding mode is also important for NF449 potency (e.g. K136, N184, Q231, K70, T216, F188, V229, L218, R292, K68). Have the authors considered making some of these mutations (I am aware that some are necessary for the action of ATP, so would render the channel non-functional, but is that the case for all of them)?

Response: We thank the reviewer for these comments and agree that supramolecular binding of NF449 is surprising. We have now done a significant amount of mutational work to show the importance of key residues necessary for NF449 binding. We have added this data in a new figure (Figure 6), where we break the mutational analysis into residues critical for the "inner" NF449 and residues critical to the "outer" NF449. While some of the residues do affect the action of ATP, we identified at least one distinct residue critical for the inner NF449 and a different residue critical for the outer NF449, neither of which affect ATP binding. This allows the very clear conclusion that residues that interact with both the "inner" and "outer" NF449 molecules are important for the high-affinity antagonism observed for NF449, consistent with our structural results.

(2) Results, line 312. Is there any effect on ATP potency of the R139A mutation? If this mutation affects ATP potency, it will complicate the interpretation of the NF449 IC₅₀ data.

Response: This is an excellent control experiment. The R139A mutation has no effect on ATP potency (EF₅₀) but a profound effect on the inhibition potency (IC₅₀) of NF449 (please see the newly added Figure 6). This allows us to make the firm conclusion that residue R139 is critical for NF449's high-affinity binding and inhibition.

(3) Results, line 312. Could the dual-NF449 bound structure be an artefact of incubation with a high concentration of antagonist for structure determination?

Response: Thank you for this question. We added ~400 μ M NF449 to ~100 μ M concentration of protomer (5 mg/mL). This is only a 4:1 molar excess of ligand over the protomer concentration and only 2:1 excess over available binding sites. This is not excessively high at all, and we would not expect this ratio of ligand to binding site to result in artifactual binding. To this point, the well-resolved density of the two bound NF449 molecules provides strong evidence that the dual-binding mode is specific. Artifact density from excessive antagonist, which would presumably bind the receptor non-specifically, should not result in clear density due to rotational averaging in data processing. Our structure also highlights residue interactions specific to either molecule of NF449 (the "inner" and the "outer") and we now confirm both are important with electrophysiology experiments, providing a strong basis for a dual-ligand binding mode. The most important evidence is the mutation of residue R139 which reduces the apparent inhibitory potency of NF449 without affecting the EC₅₀ of ATP. Since R139 does not form interactions with the inner NF449, the presence of an outer binding site is the most likely explanation for why mutation of this residue has such a profound effect on the IC₅₀.

Minor comments

(1) Nomenclature. The modern naming convention for P2X receptors is not to use a subscript for the receptor subtype number (should be P2X1) and this should be followed throughout.

Response: Thank you for pointing this out. We have now converted to the standardized P2X receptor naming convention in the manuscript.

(2) Introduction, first two paragraphs. It would be more succinct to briefly state the physiological roles of P2X1 and why it is a potential therapeutic target, and that a molecular understanding of antagonism is lacking, as the work presented does not contain any experimental data relating to platelet biology.

Response: We are really hoping to keep the introduction as it is. We feel it nicely highlights the physiological role of the P2X1 receptor and provides points of interest to the more translationally minded audience members.

(3) Introduction, paragraph 3. It may be factually accurate to state that 'it is not known ... why P2X7 receptor is the least sensitive to ATP' but it has been suggested that P2X7 can only bind ATP⁴⁻, and that there are differences in the structure surrounding the ATP binding site that can account for the relatively low sensitivity to ATP in this subtype.

Response: Thank you for the comment. We have changed this to now say "not fully understood" (line 66).

(4) Introduction, line 100. This should state 'will facilitate' rather than 'facilitates'.

Response: Thank you for pointing out this error, which we have now fixed (line 91).

(5) Results, line 143. It would be better to move the statement in lines 164-167 to the start of this section.

Response: Thank you for the suggestion. We have now moved this statement to the start of the corresponding paragraph (starting at line 134).

(6) Results, line 193. How do you know that the structure you observe is the desensitized state? Is it by analogy to the hP2X3 structure, or is it an assumption because P2X1 desensitizes very quickly with ATP bound? I do not dispute that the authors are observing the desensitized state, but I think they should qualify this at the start of the section describing the structure.

Response: Thank you for these questions. We have now specifically highlighted why we define the structure as the desensitized state and moved this explanation to the beginning of the section (lines 191-192).

(7) Results, lines 282-291. Do the authors have any direct evidence that it is Mg-ATP rather than Na-ATP bound to the receptor? Can they elaborate on the CheckMyMetal results? I agree that it is more likely that it is Mg-ATP, but the authors should be clear about what is direct evidence and what is assumption.

Response: Thank you for the comments. We have now made it clear that we chose to model the density as a Mg²⁺ because it is more likely considering the endogenous source of ATP in the ATP-bound desensitized state structure and that the CheckMyMetal results indicate a Mg²⁺ ion is consistent with the data (but this data does not prove it is a Mg²⁺). This is discussed in lines 290-296.

REVIEWERS' COMMENTS

Reviewer #1 (Remarks to the Author):

The authors responded to the comments well by including extra experimental data and deeper discussions. This is a good paper and I have no further comments.

Reviewer #2 (Remarks to the Author):

I thank the authors for addressing my comments.